# ibicus: a new open-source Python package and comprehensive interface for statistical bias adjustment and evaluation in climate modelling (v1.0.1)

Fiona Spuler[1,*], Jakob Wessel[2,*], Edward Comyn-Platt[3], James Varndell[3], and Chiara Cagnazzo[3]

[1]Department of Meteorology, University of Reading
[2]Department of Mathematics and Statistics, University of Exeter
[3]European Centre for Medium-Range Weather Forecasts (ECMWF)
[*]These authors contributed equally to this work.

**Correspondence:** Jakob Wessel (jw1301@exeter.ac.uk)

**Abstract.** Statistical bias adjustment is commonly applied to climate models before using their results in impact studies. However, different methods, based on a distributional mapping between observational and model data, can change the simulated trends, as well as the spatiotemporal and inter-variable consistency of the model, and are prone to misuse if not evaluated thoroughly. Despite the importance of these fundamental issues, researchers who apply bias adjustment currently do not have the tools at hand to compare different methods or evaluate the results sufficiently to detect possible distortions. Because of this, widespread practice in statistical bias adjustment is not aligned with recommendations from the academic literature. To address the practical issues impeding this, we introduce *ibicus*, an open-source Python package for the implementation of eight different peer-reviewed and widely used bias adjustment methods in a common framework and their comprehensive evaluation. The evaluation framework introduced in *ibicus* allows the user to analyse changes to the marginal, spatiotemporal and inter-variable structure of user-defined climate indices and distributional properties, as well as any alteration of the climate change trend simulated in the model. Applying *ibicus* in a case study over the Mediterranean region using seven CMIP6 global circulation models, this study finds that the most appropriate bias adjustment method depends on the variable and impact studied and that even methods that aim to preserve the climate change trend can modify it. These findings highlight the importance of a use-case-specific choice of method and the need for a rigorous evaluation of results when applying statistical bias adjustment.

## 1 Introduction

Even though climate models have greatly improved in recent decades, simulations of present-day climate of both global and regional climate models still exhibit biases Vautard et al. (2021). This means that there are systematic discrepancies between statistics of the model output and observational distribution Maraun (2016). These discrepancies in the two distributions become especially relevant when using the output of climate models for local impact studies that often require focus on specific threshold metrics such as dry days, for example when running hydrological Hagemann et al. (2011) or crop models Galmarini et al. (2019).

To account for and potentially correct these biases, it has become common practice to post-process climate models using statistical bias adjustment before using their output for impact studies. The idea behind statistical bias adjustment is to calibrate a statistical transfer function between the observed and climate model distribution of a chosen variable. A variety of statistical bias adjustment methods have been developed and published in recent years, ranging from simple adjustments to the mean, to trend-preserving adjustments by quantile and further multivariate adjustments (Michelangeli et al., 2009; Li et al., 2010; Cannon et al., 2015; Vrac and Friederichs, 2015; Maraun, 2016; Switanek et al., 2017; Lange, 2019, and more). While this paper focuses primarily on methods that are applied at each grid cell individually, the use of multivariate methods is further discussed in section 5.

Despite widespread use both within the scientific community (see, for example, IPCC, 2021, 2022), as well as by climate service providers and practitioners (see, for example, climate scenarios used by central banks across the world, NGFS, 2021), bias adjustment is known to suffer from fundamental issues. These issues have been highlighted, among others, by Maraun et al. (2017) who show that bias adjustment not only has limited potential to correct misrepresented physical processes in the climate model but can also introduce new artefacts and destroy the spatiotemporal and inter-variable consistency of the climate model. To avoid misuse, Maraun et al. (2017) recommend the evaluation of non-calibrated aspects, the development of process-informed bias adjustment methods based on an understanding of climate model errors, and the selection of climate models that represent the large-scale patterns and feedback relevant to the impact sufficiently well.

We argue that the remedies mentioned above are not common practice due to practical issues with statistical bias adjustment. As Ehret et al. (2012); Maraun (2016); Casanueva et al. (2020) highlight, different bias adjustment approaches are appropriate for different use cases. However, methods that exist in the academic literature are published either only as papers, bias adjusted datasets (Dumitrescu et al., 2020; Mishra et al., 2020; Navarro-Racines et al., 2020; Xu et al., 2021, and more) or as stand-alone packages across multiple programming languages (Iturbide et al., 2019; Lange, 2021b; Michelangeli, 2021; Cannon, 2023, and more), often without accompanying evaluation or evaluation frameworks. This gives users who are not necessarily experts in these methods limited options to choose the bias adjustment method most appropriate for their use case and evaluate the results sufficiently to detect issues.

In this paper, we introduce *ibicus*, an open-source Python package for the implementation, comparison and evaluation of bias adjustment for climate model outputs. The contribution of *ibicus* is two-fold: For one, it introduces a unique unified interface to apply eight different peer-reviewed and widely used bias adjustment methodologies. The implemented methods include Scaled Distribution Mapping (Switanek et al., 2017), CDFt (Michelangeli et al., 2009), Quantile Delta Mapping (Cannon et al., 2015) and ISIMIP3BASD (Lange, 2019). Further, it develops an evaluation framework for assessing distributional properties and user-defined climate indices (covering but not limited to the ETCCDI indices – Zhang et al., 2011) not only along marginal but also temporal, spatial and multivariate dimensions. Applying *ibicus* in a case study over the Mediterranean region, we find that the most appropriate method indeed depends on the variable and impact studied and that the evaluation of spatiotemporal metrics can identify issues with bias adjustment that would not be found when only marginal, i.e. calibrated aspects are evaluated. Further, we find that even methods that aim to preserve the trend of the climate model can modify it, and that bias adjustment modifies the overall climate model ensemble spread.

The remainder of this paper is structured as follows. Section 2 gives an introduction to statistical bias correction methodologies and section 3 presents *ibicus*, covering both the details of different bias adjustment methodologies and evaluation metrics implemented, as well as the software design of the package. In section 4, we present the results of the case study before drawing conclusions in section 5.

## 2    Background

### 2.1    Statistical bias adjustment of climate models

Climate model biases can be defined as "systematic difference between a simulated climate statistic and the corresponding real-world climate statistic" (Maraun, 2016). These biases mostly stem from the imperfect representation of physical processes such as orographic drag, convection, or land-atmosphere interactions. This leads to the incorrect representation of features such as the mean and variance of observed temperature or the spatial properties of extreme rainfall over a certain area.

Bias adjustment methods for climate models have their origin in methods developed for the post-processing of Numerical Weather Prediction (NWP) models. The rationale is to calibrate an statistical transfer function between model simulations and observations over the historical period, that is then applied to the model simulation for the period of interest, often in the future. However, in contrast to NWP models, there is no direct correspondence between the time series of observations and the climate model in historical simulations. This means that typical regression-based approaches used for NWP are not applicable. Rather, properties of the statistical distribution of the two variables, such as the mean or quantiles, are mapped to each other when bias adjusting climate models. Furthermore, the magnitude of biases in climate models can be much larger, whereas NWP forecasts are tightly constrained by recent observations.

The most common approaches to the bias adjustment of climate models include a simple adjustment of the mean (Linear Scaling), a mapping of the two entire cumulative distribution functions (Quantile Mapping), or more advanced methods that also aim to preserve the trend projected in the climate model (such as CDFt or ISIMIP3BASD). Most of these methods, however, should rather be seen as method families that have some core characteristics - quantile mapping, for example, always implements a correction in all quantiles - as well as some interchangeable components, such as their handling of dry days, that they might share with other methods. The distinction between core characteristics and interchangeable components varies from method to method, as will be discussed in more detail in the description of the software package. An alternative approach, often termed Delta Change method, adjusts the historical observations to incorporate the climate model trend (see, for example, Olsson et al., 2009; Willems and Vrac, 2011; Maraun, 2016). The practice of using bias adjustment methods to also downscale the climate model has been criticised in various publications (von Storch, 1999; Maraun, 2013; Switanek et al., 2022), therefore this paper focuses on bias adjustment of climate models purely for the purpose of reducing biases at constant resolution.

The use of bias adjustment methods has become standard practice in academic climate impact studies, and increasingly outside of academia in national assessment reports or other climate services. For example, the ISIMIP3BASD methodology (Lange, 2019) is implemented as the only bias adjustment method as a standard pre-processing step in the Inter-Sectoral Impact Model Intercomparison Project (ISIMIP) impact modelling framework that is used in the climate risk scenarios published by

central banks (NGFS, 2021). However, applying statistical bias adjustment to climate models raises a number of important considerations and issues which we categorize into *fundamental* and *practical* issues for the purpose of this paper.

## 2.2 Fundamental issues with statistical bias adjustment and evaluation

Climate model biases in statistics at the grid-cell level can stem from larger-scale biases of the model such as biases in larger drivers such as El Niño, the lack of local feedback to these drivers or the misplacement of storm tracks in a region. However, univariate statistical bias adjustment methods are only as capable as their assumptions and input data and therefore correct only the impact these larger-scale biases have on the distribution of the variables at grid cell level (Maraun et al., 2017).

Univariate bias adjustment might also deteriorate the spatial, temporal or multivariate structure of the climate model. This is particularly problematic for compound events which have been argued to be of particularly high societal relevance due to their elevated impacts and neglect in standard extreme event evaluation approaches (Zscheischler et al., 2018, 2020). As this issue will not be detected in location-wise cross-validation approaches, it is necessary to evaluate bias adjusted data with a particular focus on spatial, temporal and multi-variable components (Maraun et al., 2017; Maraun and Widmann, 2018a).

Furthermore, bias adjustment can modify the climate change trend simulated by the model, in particular, that of threshold-sensitive climate indices such as dry days (Dosio, 2016; Casanueva et al., 2020). This holds in general for non-trend-preserving methods, but can also be the case for any trend-preserving methods such as ISIMIP3BASD. Reasons for the modification of the trend by 'trend-preserving' methods can be traced to the underlying statistical method and assumptions, such as the specific treatment of values between a variable bound and another threshold, or parametric and non-parametric distribution fits used in different stages of the bias adjustment.

To justify any kind of trend modification by the bias adjustment method, it is necessary to make an assumption about how present-day bias relates to biases in the future period (Christensen et al., 2008). This can be based on the assumption that climate model biases are stationary in time (Gobiet et al., 2015): for example, based on this assumption, Ivanov et al. (2018) developed a theoretical model to justify future trend modifications by the bias adjustment method based on present-day biases. However, Chen et al. (2015); Hui et al. (2019), show that while temperature biases can be approximated as stationary, precipitation biases cannot. Similarly, Van de Velde et al. (2022) show a clear impact of non-stationarity on bias adjustment, in particular for precipitation. Trend-preserving bias adjustment methods on the other hand assume, at least to some degree, that the raw climate model trend constitutes our best available knowledge for subsequent impact studies. In line with this, Maraun et al. (2017) argue that the modification of the trend of a climate model based purely on statistical reasoning is not defendable, and should, rather be based on physical process understanding and reasoning about the large-scale drivers involved.

There are some options available to cope with these fundamental issues in impact studies: the first is to discard climate models that misrepresent large-scale circulation relevant to the problem at hand. The second is to conduct a careful evaluation of multivariate aspects of the bias adjusted climate model to identify potential artefacts and discard methods that introduce these before proceeding with the impact study. The third is to develop process-informed multivariate bias adjustment methods that for example include large-scale covariates such as weather patterns (Maraun et al., 2017; Verfaillie et al., 2017; Manzanas and Gutiérrez, 2019). These more elaborate methods require an even more careful case-by-case model selection and evaluation.

## 2.3   Practical issues with bias adjustment and the availability of open-source software

Addressing these fundamental issues and improving the application of bias adjustment is impeded by a number of practical issues.

The first practical issue is that the comparison of different bias adjustment methods and their adaptation to a specific application is not easily possible for a user. This is because the code to implement different methodologies is published, if at all, across different software packages and languages, impeding interoperability. Users also have the option of downloading already bias adjusted datasets which improves ease of access but does not allow for any custom adjustments (Dobor et al., 2015; Famien et al., 2018; Dumitrescu et al., 2020; Xu et al., 2021). The second practical issue is that available software packages are not accompanied by evaluation methods beyond marginal aspects. As the evaluation of bias adjustment is not straightforward, this makes it difficult for a user to detect artefacts or identify improper results by assessing multivariate properties of the climate model, rendering bias adjustment prone to misuse (Maraun et al., 2017).

These practical issues jeopardize the current implementation of statistical bias adjustment. Addressing these issues does not solve the more fundamental issues but can improve common practice and enhance transparency.

An example of good practice is the MIdAS package which introduces a new bias adjustment method that is compared to other methods in Berg et al. (2022). However, even though the package is in principle extendable, other methods are not implemented in practice, nor is an adjustable evaluation framework developed.

## 3   ibicus – an open-source software package for bias adjustment

To address the practical issues outlined in the previous section we introduce *ibicus*, an open-source Python package for the bias adjustment of climate models and evaluation thereof. *ibicus* introduces a unified, modular, software architecture within which eight state-of-the-art peer-reviewed and widely used bias adjustment methodologies are implemented. This enables researchers to apply different methods through a common interface, and modify components of the methods, such as the treatment of dry days, based on region and impact of interest. The code implementation of each methodology is based on the cited academic publication, as well as available accompanying code that was re-written and modularised to fit the developed interface. Consistency with the original implementation was ensured through rigorous testing and correspondence with the authors of the different methodologies. The package provides an extensive evaluation framework covering spatial, temporal and multivariate aspects. As part of this, we develop a generalized threshold metric class that allows the user to evaluate both frequently used climate metrics such as frost days or dry days, as well as define their own threshold metrics targeted to the specific impact study. The spatiotemporal evaluation of threshold metrics enables the user to detect artefacts and evaluate compound events before and after bias adjustment. *ibicus* is designed to be flexible and easy to use, facilitating both the "off the shelf" use of methods as well as their customization and allowing use in notebook environments all the way up to the integration with high-performance computing (HPC) packages such as dask (Rocklin, 2015). This section provides an overview of the key features of *ibicus*. A more complete user guide and tutorials can be found on the documentation page of the package.

## 3.1 Data input

Bias adjustment requires observational data and climate model simulations during the same historical period and climate model simulation for the (future) period of interest. *ibicus* operates on a numerical level, taking three-dimensional (time, latitude, longitude) numpy arrays as input and returning arrays of the same shape and type. This choice was made to ensure interoperability with different geoscientific computing packages such as xarray (Hoyer and Hamman, 2017) or iris (Met Office, 2010), as well as operation in different computing environments and integration with dask (Rocklin, 2015).

## 3.2 Bias adjustment

*ibicus* represents each bias adjustment methodology as a class which inherits generic functionalities from a base 'debiaser' class, such as the common initialization interface and a function applying the 'debiaser' in parallel over a grid of locations. The base 'debiaser' class makes the package easily extendable, as a new bias adjustment methodology can inherit these generic functionalities and requires only the specification of a function which applies the methodology for a given location ('apply_location').

Each 'debiaser' object is initialized separately for each variable and requires several class parameters. These are specific to the bias adjustment methodology and include parameters such as the distribution used for a parametric fit or the type of trend preservation applied. For a number of methodology-variable combinations, default settings exist that are described in the documentation. Default settings are labeled 'experimental' if they have not been published in the peer-reviewed literature but are proposed by the package authors after extensive evaluation. It is possible and encouraged to modify the parameters even when default settings exist to adapt the method to a given use case. For example, if precipitation extremes are of special interest, the user could choose to modify the parametric fit for this variable as the gamma distribution – an often used default – might underestimate precipitation extremes (Katz et al., 2002). After initialization, each debiaser object has an 'apply' method to apply bias adjustment to climate model data. This takes a 3-dimensional numpy array of observations, as well as historical and future climate model simulations as input, together with optional date information for running windows. The apply function can be run in parallel to speed up execution and integrates with dask for deployment in HPC environments.

Table A1 provides an overview of the methodologies currently implemented in ibicus, chosen to cover some of the most widely used bias adjustment methods in current practice. These methods are based on different assumptions, making them suitable for different applications. For example, ISIMIP3BASD is a parametric trend preserving quantile mapping which might be appropriate if the variable approximately follows a known parametric structure and the climate change trend in all quantiles is judged to be realistic. If these assumptions are not valid, a non-parametric method such as CDFt or a non-trend preserving method such as Quantile Mapping might be more appropriate. Alternatively, if changes in extremes are of special interest, a parametric method based on extreme value theory might be adequate. As noted in the background section, different methods should rather be viewed as method families that have core characteristics and interchangeable components in their ibicus implementation. An example of this is the treatment of dry days in different methods: While the treatment of dry days is entangled in the method design for SDM, CDFt and ISIMIP and cannot be changed by the user, QM methods allow for different

**Table 1.** Distinctions between different bias adjustment methods and important considerations motivating the choice.

| Statistic / quantiles | Methods for bias adjustment range from simple adjustments to the mean (Linear Scaling – LS) or mean and variance (LS) to adjustments to all quantiles of the distribution. |
|---|---|
| Parametric or non-parametric Method | Non-parametric methods are restricted to the range of observed/modelled data in their "historical period" and might not handle extremes well, while parametric methods introduce additional assumptions. *ibicus* allows users to implement all methods non-parametrically by modifying method attributes. Based on the default arguments, QM, QDM, ECDFM and SDM are parametric methods while CDFt is non-parametric and ISIMIP3BASD is semi-parametric. For each method using a parametric distribution, it is possible to exchange it with a different one. |
| Time-window | Some methods include a running window to calculate different transfer functions depending on seasonality (QDM, ISIMIP3BASD, CDFt is applied by month) whilst others do not account for seasonality explicitly. |
| Trend-preservation and stationarity assumption | Methods such as quantile mapping can modify the trend in the climate model. This might be sensible if the trends are taken to be unrealistic and related to present-day biases, as discussed in the background section (Boberg and Christensen, 2012; Gobiet et al., 2015; Doblas-Reyes et al., 2021). However, in other cases, the trend might be considered credible and should be preserved. Methods can be designed to preserve trends in the mean (DC, LS, dQM), mean and variance (dQM) or all quantiles (CDFt, ECDFM, QDM, ISIMIP3BASD, SDM) - although even then they are not guaranteed to do so. Often trends are distinguished between additive trends (as for temperature) and multiplicative trends (as for precipitation where trends in intensity occur), however not all methods share this distinction. The question of trend preservation is related to the assumption made that the bias is 'stationary', as mentioned in the background section. The assumption is explicitly made by Quantile Mapping. SDM explicitly relaxes the assumption, CDFt and QDM account for it by including a running window over the future period in addition to one over the year. |
| Treatment of dry days and extremes | Methods have different ways of handling certain aspects of the distribution such as extreme values or dry days in the case of precipitation. For extremes some methods use an extrapolation based on parametric distribution, which can be modified by the user for example should a mapping based on extreme value theory be required. For dry days the ISIMIP, SDM and CDFt methods provide an explicit handling that might be appropriate in some situations but not in others. QDM treats the mapping of dry days as a censoring problem and adjusts them together with the body of the distribution whilst for methods like QM and ECDFM the user has the choice of different treatment methods. |

treatment of dry days depending on the use-case. Table 1 highlights further methodological considerations differentiating different method families. A detailed description of each individual component of each method is beyond the scope of this paper but can be found in the detailed ibicus software documentation provided online.

### 3.3 Evaluation

Physical consistency in space, time or between variables is not ensured when using univariate bias adjustment methods. Furthermore, the trend of the climate model might be modified, and the bias of some statistics or impact metrics might be increased through some bias adjustment methods – even if it is removed in certain quantiles. The *ibicus* evaluation framework offers a

**Table 2.** Attributes of the threshold metrics class.

| Threshold Attribute | Description |
|---|---|
| Name | Name of the threshold |
| Value(s) | Values defining the threshold (to compare climate model or observations against). |
| Description | Brief description of the threshold. |
| Type | Whether values shall fall above, below, outside or between threshold(s). |
| Scope | Whether the threshold(s) is defined daily, monthly, seasonally, or overall (different for each time category, or not). |
| Locality | Whether the threshold is defined location-wise or globally (different at each location or not). |

collection of tools to identify these issues and compare the performance of different bias adjustment methods for variables of interest, building on previous efforts such as the VALUE evaluation framework for statistical downscaling (Maraun et al., 2019).

### 3.3.1 Metrics and design

The evaluation framework consists of two components: 1) the evaluation of bias adjustment on a validation/testing period that enables comparison of the bias adjusted model with observations, and 2) the analysis of trend preservation between the validation and future, or any two future periods. The latter component is necessary as bias adjustment methods can modify the climate change trend, even with methods that are designed to preserve it, as demonstrated by the case study in section 4. In the absence of evidence to the contrary, trend-preserving methods should be preferred as statistical bias adjustment methods usually do not have an underlying physical reasoning for modifying a particular trend.

In both components of the evaluation framework, there are two kinds of metrics that can be evaluated using *ibicus*, termed statistical properties and threshold metrics. Statistical properties allow the user to compare properties of the observational distribution and the climate model distribution - such as the mean or different quantiles - before and after bias adjustment. Threshold-based climate indicators are often of special interest for climate impact studies – for example, frost days, by time of year, could be of interest for agricultural or biodiversity impacts – and where the success of bias adjustment methods is particularly desirable (Dosio et al., 2012; Dosio, 2016). A number of threshold metrics are implemented by default in the package. A new threshold metric can be specified by the user along the dimensions in table 2. Accumulations such as monthly total precipitation can also be estimated. Using these definitions, the evaluation module covers but is not limited to the indices developed by the Expert Team on Climate Change Detection and Indices (ETCCDI - Zhang et al., 2011) used in many application studies.

Since location-wise evaluation is not sufficient to decide whether a bias adjustment method is fit for the use-case, the module offers the functionality to evaluate location-wise, as well as spatiotemporal and multivariate metrics both in terms of threshold metrics and statistical properties. The table 3 gives an overview of the implemented methods.

**Table 3.** Overview of evaluation categories implemented in ibicus.

| | Statistical Properties | Threshold Metrics |
|---|---|---|
| Location-wise | **Calculation**: location-wise bias (absolute and percentage) in different distributional properties (quantile, mean) of climate model before and after applying different bias adjustment methods.<br>**Visualization**: boxplot across locations and spatial plot. Plotting functions for visual inspection of observed and climate model distribution (histogram and CDF). | **Calculation**: location-wise bias (absolute in days/year and percentage) in the frequency of singular threshold exceedance events in climate model before and after bias adjustment methods.<br>**Visualization**: boxplot across locations and spatial plot. |
| Temporal | - | **Calculation**: distribution of spell lengths of threshold exceedances (for example dry spell length).<br>**Visualization**: plot of empirical CDF. |
| Spatial | RMSE of between spatial correlation matrices at each location. | **Calculation**: distribution of spatial cluster size of threshold exceedances (for example spatial size of heatwaves).<br>**Visualization**: plot of empirical CDF. |
| Spatio-temporal | - | **Calculation**: distribution of spatiotemporal cluster size of threshold exceedances (for example spatiotemporal extent of heatwaves).<br>**Visualization**: plot of empirical CDF. |
| Multivariate | **Calculation**: correlations between chosen pair of variables at each location.<br>**Visualization**: spatial plot. | **Calculation**: conditional probability of threshold co-occurrence (such as dry and hot days) in observations and climate model before and after bias adjustment.<br>**Visualization**: boxplot. |
| Trend | **Calculation**: location-wise bias in the multiplicative or additive trend of a threshold metric or distributional property (mean, quantiles) – percentage change between climate model before and after bias adjustment.<br>**Visualization**: boxplot across locations and spatial plot. | |

Finally, different bias adjustment methods rely on different assumptions such as certain parametric distributions providing suitable fits. The evaluation framework includes functions to assess the fit of parametric distributions and the seasonality of the variable to help the user make decisions on how to customize the bias adjustment method to their application.

# 4   Implementation of ibicus in the Mediterranean region

We demonstrate the comparison and evaluation of different bias adjustment methods by applying *ibicus* over the Mediterranean. Rather than conducting a comprehensive evaluation for a single use case, our aim is to highlight the use-case dependency of the method choice more broadly and hence the necessity of targeted evaluation beyond marginal aspects. We, therefore, choose to

limit this case study to the bias adjustment of global climate models, even though specific impact studies often but not always (IPCC, 2021) use higher-resolution models over the target region.

## 4.1 Data and Methods

We consider the Mediterranean region, between 35-45°N latitude and 18°W to 45°E longitude and apply bias adjustment to seven Coupled Model Intercomparison Project Phase 6 (CMIP6) models, selected based on the use in previous studies in the Mediterranean region (Zappa and Shepherd, 2017; Babaousmail et al., 2022). The chosen models include ACCESS-CM2, CMCC-ESM2, IPSL-CM6A-LR, MIROC6, MPI-ESM1-2-LR, MRI-ESM2-0 and NORESM2-MM. Table B1 in the appendix provides more details on these models. We used the historical runs as well as the SSP5-8.5 experiments. We compare four

widely used bias adjustment methods that are implemented in *ibicus*: ISIMIP3BASD (Lange, 2019), applied amongst others by Jägermeyr et al. (2021); Pokhrel et al. (2021) as well as impact models run under the ISIMIP framework), Scaled Distribution Mapping (Switanek et al. (2017), applied amongst others as pre-processing step to assess changes in high impact weather events over the UK in Hanlon et al. (2021)), as well as Quantile Mapping (applied in impact studies such as Babaousmail et al., 2022) and Linear Scaling as reference methods. These four methods are applied to daily total precipitation (pr) and

daily minimum near-surface air temperature (tasmin), chosen to cover two different types of variables (bounded vs unbounded, different distributions etc) that are both highly relevant for many impact studies. The bias adjustment methods are used with their ibicus default settings for both variables (for more details see table A1 and the software documentation). This means that the ISIMIP and SDM methods provide an explicit adjustment of dry day frequencies, whilst for QM they are treated as censored and the method based on Cannon et al. (2015) is applied and LS provides no explicit adjustment, scaling all values.

We use ERA5 reanalysis data (Hersbach et al., 2020) as an observational reference, conservatively regridded to match the resolution of the selected climate models. The historical data ranges from January 1st, 1959 to December 31st, 2005, with the data from January 1st 1959 to December 31st 1989 serving as the historical/reference period and used as a training dataset and the subsequent period: January 1st 1990 to December 31st 2005 used for validation purposes. Bias adjustment is applied to the validation period as well as the future period: January 1st 2080 to December 31st 2100,

We demonstrate four bespoke impact metrics related to daily minimum temperature and daily total precipitation, defined using the *ibicus* threshold metrics class.

  – tasmin < 10°C (283.15K) which was chosen based on Droulia and Charalampopoulos (2022) who estimate climate impacts to viniculture noting that above >10°C grapevines are in their optimal photosynthesis zone.

  – tasmin greater than the seasonal 95th percentile of the daily minimum temperature in each grid cell during the historical
period (1959- 1989). This can be an indicator of the impacts of heatwaves (Raei et al., 2018).

  – Dry days (daily precipitation <1mm) and very wet days (daily precipitation >10mm) as two ETCCDI indices.

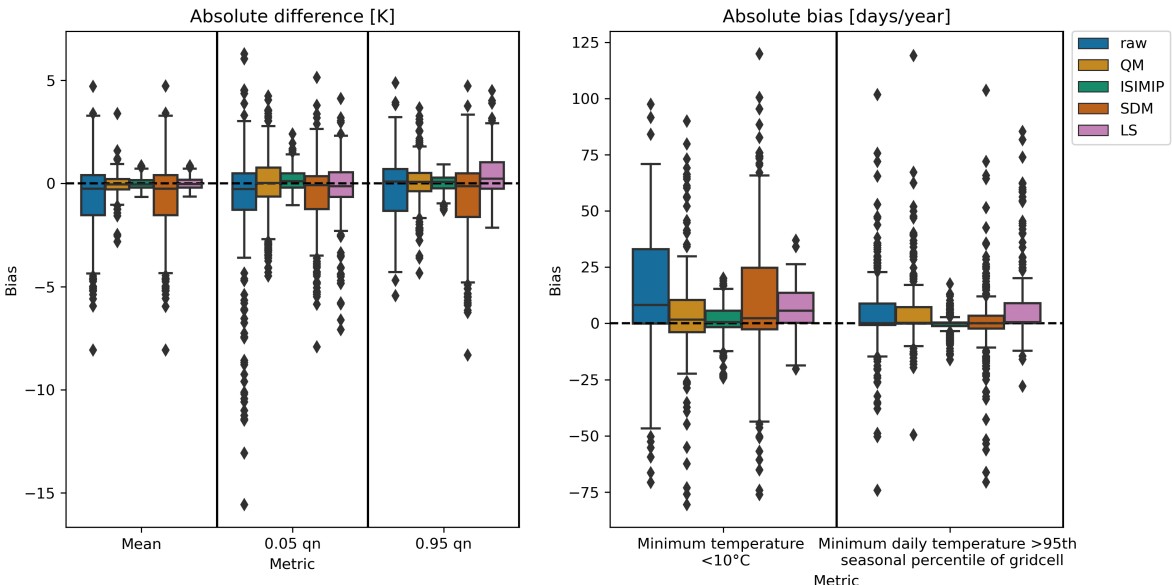

**Figure 1.** Distribution across locations of marginal minimum daily temperature bias of the ACCESS-CM2 climate model before bias adjustment (raw) and after applying the ISIMIP3BASD bias adjustment method (ISIMIP), Quantile Mapping (QM), Scaled Distribution Mapping (SDM) and Linear Scaling (LS). The left panel displays the distribution of the absolute bias (in Kelvin) in the mean and 0.05 and 0.95 quantiles. The right panel displays the distribution of the absolute bias in the threshold metrics: minimum daily temperature below 10°C and minimum daily temperature above the 95th seasonal percentile defined for this grid cell, both in units of days per year. Bias (location-wise) is defined as the difference between the metric for the (bias adjustment) climate model in the validation period and the metric for the observational data in the validation period (in each grid cell, metrics calculated in the temporal dimension). This figure shows the standard *ibicus* output distribution of location-wise bias for a set of specified statistics and threshold metrics. The boxplot shows the median, the first and third quartiles as a box, the outer range (defined as Q1 - 1.5 × IQR and Q3 + 1.5 × IQR) as whiskers, and any points beyond this as diamonds.

## 4.2  Results

### 4.2.1  Evaluation of the location-wise bias on the validation period

Figures 1-3 show the marginal bias of the climate model with respect to observations over the validation period before and after bias adjustment across locations in the study area.

We find that most methods reduce but do not eliminate the marginal bias in the mean, shown for the ACCESS-CM2 model and minimum daily temperature in figure 1, while the range of reduction is varied: ISIMIP and Linear Scaling achieve more significant reductions in the bias than Quantile Mapping or Scaled Distribution Mapping. This result also holds for extremal quantiles and threshold metrics, and we even observe a slight inflation of the raw climate model bias observed in certain instances for both Quantile Mapping and Scaled Distribution Mapping.

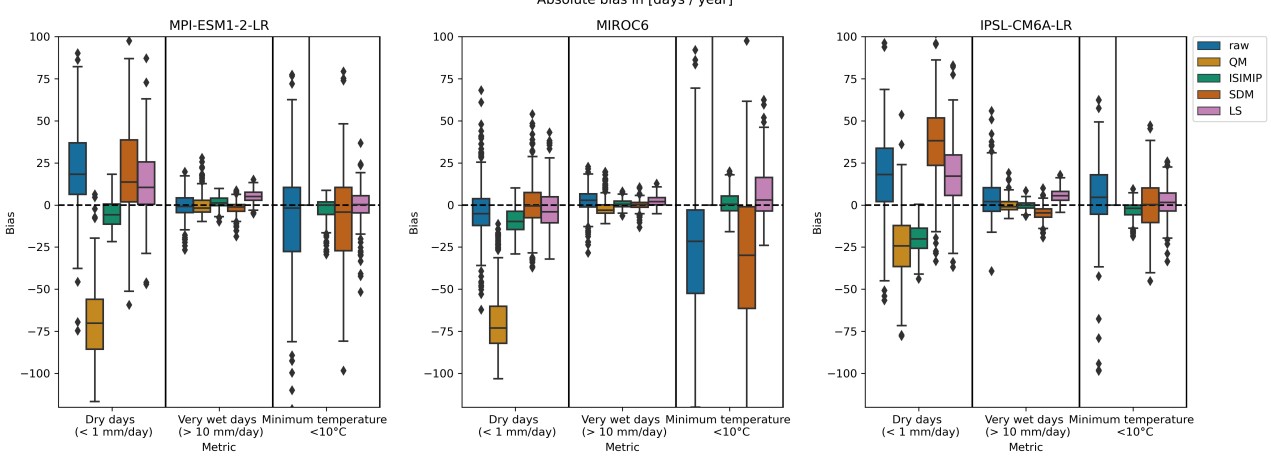

**Figure 2.** Distribution of marginal bias across locations before bias adjustment (raw) and after applying the ISIMIP3BASD bias adjustment method (ISIMIP), Quantile Mapping (QM), Scaled Distribution Mapping (SDM) and Linear Scaling (LS). Three climate models (MPI-ESM1-2-LR, MIROC6 and IPSL-CM6A-LR) and three threshold metrics (minimum daily temperature below 10°C, dry days defined as total precipitation below 1mm and very wet days defined as total precipitation above 10mm) are evaluated. The bias in minimum temperature <10°C of the climate models after applying quantile mapping is particularly large, exceeding 300%. For improved readability of the plot, we have omitted this bias adjustment - metric combination here but show the full plot in the appendix.

Furthermore, in figure 2 we see that that the success of a bias adjustment method depends on the use case, meaning the variable, metric and climate model studied. While Scaled Distribution Mapping somewhat reduces the median bias in dry days for two of the climate models, it inflates the bias in dry days for the third. On the other hand, the method reduces bias in the minimum temperature threshold metric for the IPSL-CM6A-LR model but inflates the bias in this metric for the MIROC6 model. ISIMIP3BASD on the other hand reduces the bias in dry days for the MPI-ESM1-2-LR model but increases it for the MIROC6 model. Quantile Mapping performs reasonably well for the wet-day metric but quite badly for the dry-day and minimum temperature metrics. These differences in the performance of bias adjustment methods can be due to their assumptions (a parametric distribution fit might not replicate the correct tail behaviour), and method (whether they are tailored to a specific variable or whether event frequency adjustment is implemented), as well as the physical source of the bias in the climate model.

When investigating the spatial distribution of the bias (figure 3), we find that certain methods can homogenize the spatial pattern of the bias across climate models. For example, linear scaling (LS) shifts climate models to an overestimation of very wet days in similar regions, even models like NORESM2-MM which previously underestimated these days. In other cases, methods can perform well in certain regions, but not in others. Quantile mapping (QM) seems to perform reasonably well over the Iberian peninsula, but has difficulties over Italy, especially for MPI-ESM1-2-LR where a strong underestimation is shifted

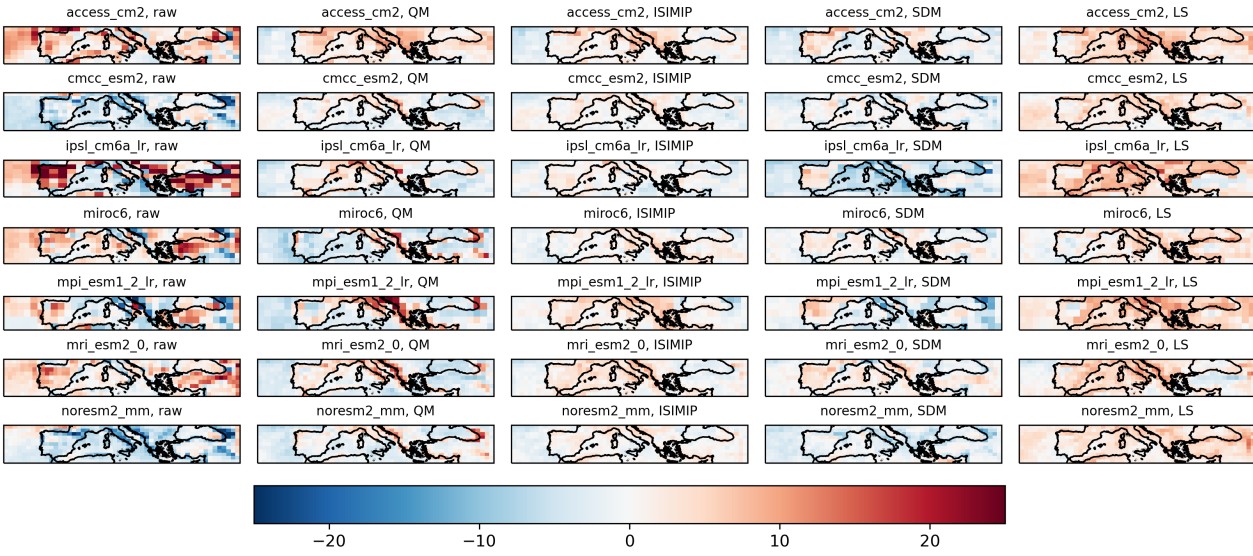

**Figure 3.** Spatial plot of marginal absolute bias in very wet days defined as total precipitation above 10mm given in [days / year]. Results are shown for seven climate models (ACCESS-CM2, CMCC-ESM2, IPSL-CM6A-LR, MIROC6, MPI-ESM1-2-LR, MRI-ESM2-0 and NORESM2-MM) before bias adjustment (raw) and after applying the ISIMIP3BASD bias adjustment method (ISIMIP), Quantile Mapping (QM), Scaled Distribution Mapping (SDM) and Linear Scaling (LS).

into a strong overestimation. This highlights the importance of investigating the spatial distribution of the marginal bias as this varies across the different regions in the Mediterranean.

### 4.2.2 Evaluation of the bias in spatiotemporal characteristics on the validation period

Moving on to the investigation of spatiotemporal characteristics, figures 4 and 5 show the cumulative distribution of spell length and spatial extent for the dry-day and minimum temperature heatwave days metric, respectively. The plots depict the standard visualization output that the *ibicus* software package produces for this type of evaluation.

The spatiotemporal characteristics investigated exhibit biases between the reanalysis data and raw climate model output. For example, it is ∼1.6 times more likely for a dry spell to exceed 20 days in the raw climate model IPSL-CM6A-LR compared to the reanalysis data.

We find that the bias in these spatiotemporal metrics can be reduced with some bias adjustment methods: for example, ISIMIP3BASD reduces the spell length bias for dry days, and Scaled Distribution Mapping reduces the bias in both spell length and spatial extent for minimum temperature heatwave days. However, this result is again inconsistent across methods and variables, and different bias adjustment methods frequently appear to increase the spatiotemporal bias: Scaled Distribution

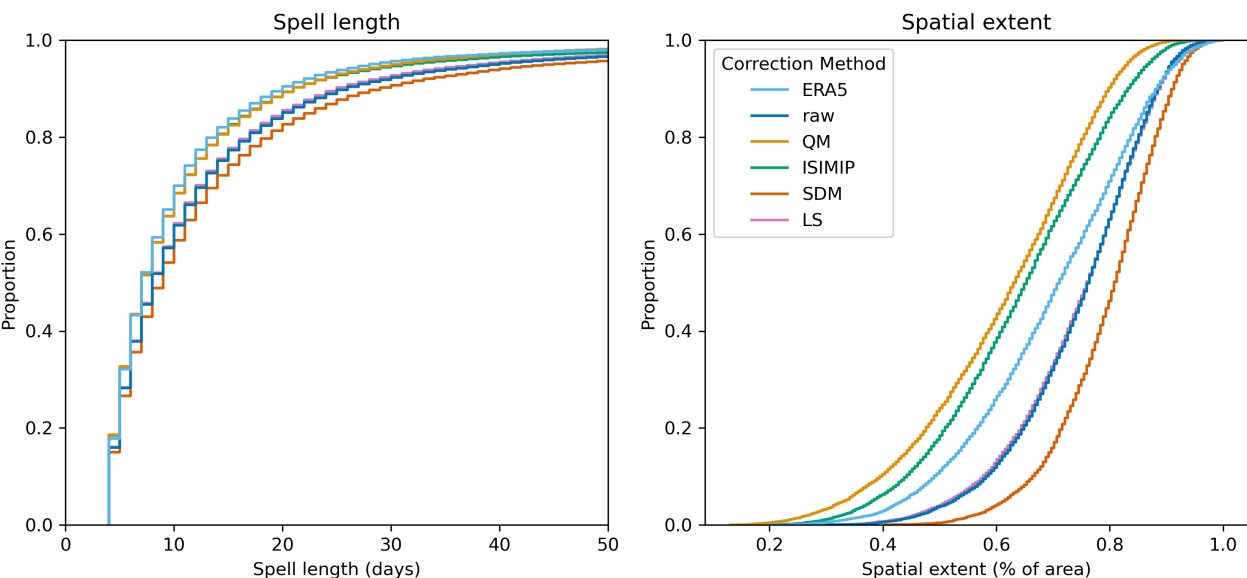

**Figure 4.** Cumulative distribution functions of spell length (left panel) and spatial extent of dry days (right panel). The spell length is defined as the length of a temporal sequence longer than three days during which a single grid cell exceeds the specified threshold. The spatial extent is defined as the fraction of cells exceeding the specified threshold, given that a single cell exceeds the threshold. This plot shows the cumulative distribution function of individual spell lengths and spatial extents at single points in time across the entire Mediterranean region in the observational data (ERA5), in the climate model IPSL-CM6A-LR before bias adjustment (raw) and after applying the ISIMIP3BASD bias adjustment method (ISIMIP), Quantile Mapping (QM), Scaled Distribution Mapping (SDM) and Linear Scaling (LS).

Mapping increases the bias in spell length and spatial extent of dry days, as do Quantile Mapping and ISIMIP3BASD when investigating the spatial extent.

These results are to some extent expected, as the selected methods are univariate methods, meaning they are calibrated location-wise and do not incorporate spatiotemporal information. However, the results highlight the need to evaluate how bias adjustment changes spatiotemporal characteristics, as these are often implicitly used in impact downstream impact studies.

### 4.2.3 Evaluation of the climate change trend before and after bias adjustment

As mentioned in the background section, the modification of the climate change signal through bias adjustment has been reported and discussed in various publications and stimulated the development of methods that aim to preserve the climate signal.

In the analysis of the dry day trend, shown in figure 6, we find that a non-trend-preserving method such as quantile mapping significantly alters the climate change trend. The axes in figure 6 were limited to +-100 for the sake of readability, however,

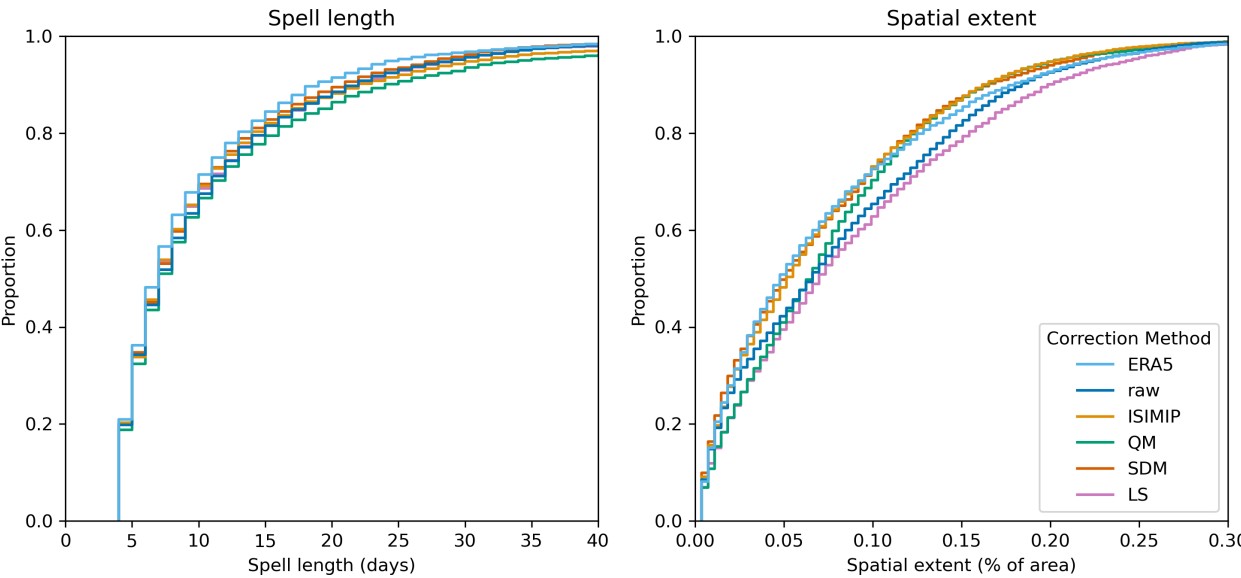

**Figure 5.** As Figure 4, but investigating the threshold of minimum daily temperature exceeding its 95th seasonal percentile defined per grid cell for the climate model ACCESS-CM2.

a limited number of data points show even larger biases after applying quantile mapping. The unrestricted version of this plot can be found in the appendix.

  We also find that methods that aim to preserve the trend such as ISIMIP3BASD or Scaled Distribution Mapping modify it up to 100% at some locations. For the ISIMIP method, this is presumably due to the fact that the 'future observations' through which the trend preservation is implemented are mapped using empirical CDFs, whereas the bias adjustment itself is

310 parametric. It has been argued that the normal distribution for temperature or the gamma distribution for precipitation might not adequately capture the tail behaviour of these variables (Katz et al., 2002; Nogaj et al., 2006; Sippel et al., 2015; Naveau et al., 2016). This is particularly relevant when investigating the trend of high or low quantiles, as well as threshold metrics that do not sit at the centre of the distribution. Additionally, for bounded variables such as precipitation, the frequency beyond two outer thresholds is adjusted separately in the ISIMIP3BASD methodology which could lead to the change in the dry day trend

shown in figure 6.

  We find a much smaller change in the trend of the mean minimum daily temperature across methods, shown in figure 7. In fact, linear scaling barely modifies the trend at all, which is to be expected since the method only subtracts the mean bias from the future and the validation period, based on the strong assumption that the bias affects the mean only and is stationary over time.

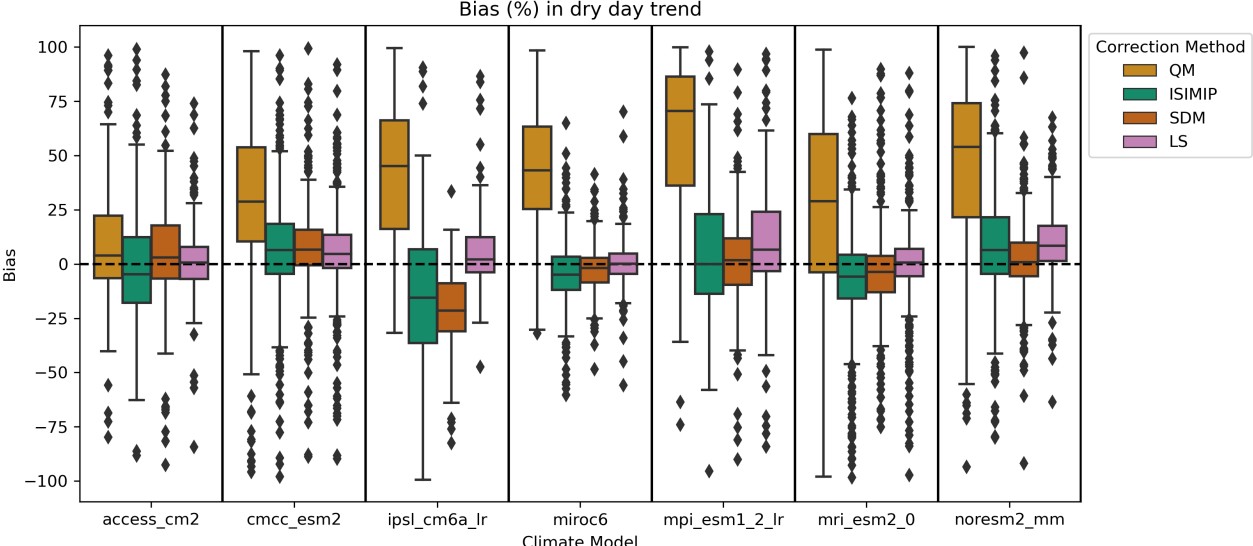

**Figure 6.** Distribution of location-wise change in the additive climate trend in dry days introduced through the bias adjustment method, computed by computing the additive trend between the validation period and the future period in both the raw and the bias adjusted model and taking the percentage difference between the two trends. The magnitude of the raw projected change in dry days depends on the climate model and, across different locations, lies between 10 fewer and 30 more dry days on average per year.

### 4.2.4 Evaluation of the variation in the climate model ensemble before and after bias adjustment

Figure 8 shows that the climate model ensemble spread of the trend of mean seasonal precipitation is modified in different ways by different bias adjustment methods which is in line with previous findings in the literature (Maraun and Widmann, 2018b; Lafferty and Sriver, 2023). Interestingly the variation (often interpreted as the uncertainty range) is not necessarily narrowed as has been postulated by some authors (Ehret et al., 2012), but even extended and shifted in some cases. From this finding, it follows that the range of uncertainty and possible worst-case scenarios analysed in subsequent impact studies might depend on the bias adjustment method used to pre-process the climate model.

The interpretation of this shift in uncertainty is related to the previously discussed questions on trend preservation, namely whether the change in the climate model trend through a statistical bias adjustment method is justified or not. This issue was mentioned by Maraun and Widmann (2018), who discuss that a minimum requirement to justify a change in the uncertainty spread through bias adjustment should be a critical evaluation of the validity of the results and the assumptions of the underlying statistical model. Given the finding in the previous section, namely that the best bias adjustment method depends on the variable, region and impact variable studied, it follows that indiscriminately applying a bias adjustment method across regions and variables without evaluation can shift the spread of the results of subsequent impact studies in a non-justified manner.

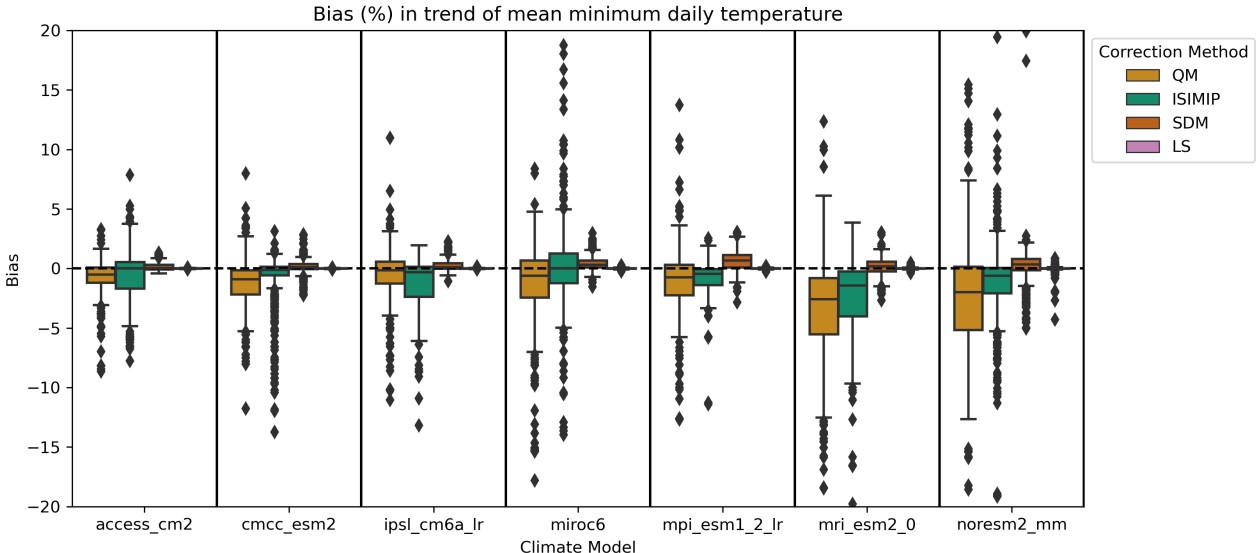

**Figure 7.** As Figure 6 but for the trend in mean minimum daily temperature. The magnitude of the raw projected change in mean minimum daily temperature again depends on the climate model and, across different locations, lies between 2-5K.

## 5 Conclusions

Statistical bias adjustment is a useful method when working with climate models to understand future climate impacts. However, there are fundamental as well as practical issues in how bias adjustment is currently used both in academic research and by practitioners in the private and government sector. One practical issue impeding good practice is the availability of open-source software to compare different bias adjustment methods and evaluate non-calibrated aspects.

This paper demonstrates that the success of a bias adjustment method depends on the variable and impact studied, and bias 340 adjustment should therefore be evaluated and adapted targeted to the region and use-case at hand. Depending on the climate model and variable of interest different methods can reduce or also increase biases by a large range, can impair or leave spatiotemporal coherence relatively unaffected. This is non-systematic across bias adjustment methods, climate models and variables/metrics of interest. Furthermore, we find that even trend-preserving methods can modify the trend in statistical properties and climate indices, and each bias adjustment method changes the climate model ensemble spread slightly differently.

With the Python package *ibicus*, we aim to provide a resource to address some of these practical issues. For one, the evaluation framework allows users to evaluate non-calibrated aspects and identify potential issues in bias adjusted data. Second, the common interface developed for different bias adjustment methods allows for a relatively easy comparison between different methods, and the selection of the method most appropriate for the use-case. Finally, the ibicus software implementation modularises certain components of different methods, such as the treatment of dry days. This allows the user to examine the

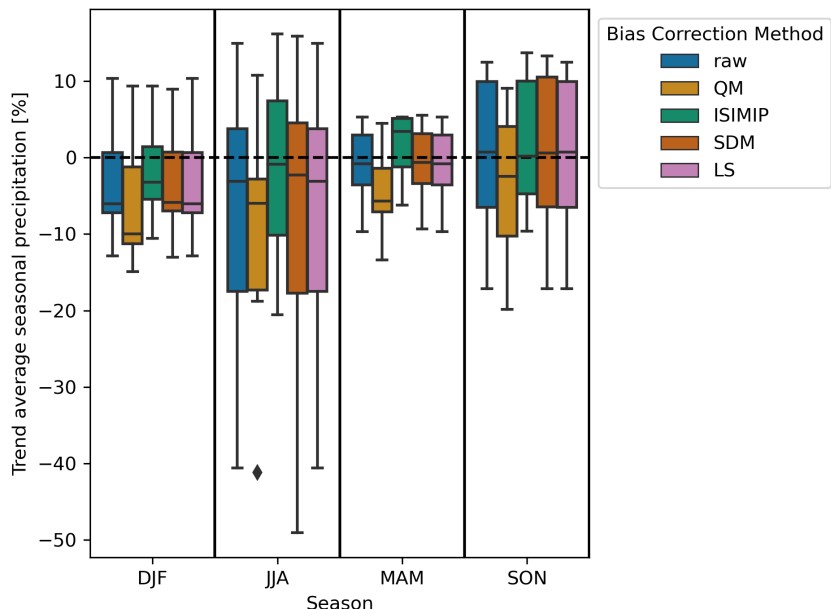

**Figure 8.** Ensemble spread of seven selected climate models (ACCESS-CM2, CMCC-ESM2, IPSL-CM6A-LR, MIROC6, MPI-ESM1-2-LR, MRI-ESM2-0 and NORESM2-MM), showing the trend in average seasonal precipitation between the validation and future period, without applying bias adjustment (raw) and after applying ISIMIP3BASD, Quantile Mapping and Scaled Distribution Mapping.

impact of detailed methodological choices for their application and select the most appropriate option, which has so far not been possible due to the dispersed implementations of different methodologies.

So far, the package implements univariate bias adjustment methods, meaning that the bias adjustment is calibrated and applied on each grid point separately. Multivariate bias adjustment methods that correct spatial, temporal, or inter-variable structures next to marginal aspects have been published, amongst others by Piani and Haerter (2012); Vrac and Friederichs

(2015); Sippel et al. (2016); Cannon (2016, 2018); Vrac (2018); François et al. (2020). We have so far chosen to focus on univariate methods as the need for careful model selection and evaluation becomes even more pertinent when using multivariate methods (Maraun et al., 2017; François et al., 2020; Van de Velde et al., 2022). Our aim was therefore to first establish a robust workflow and evaluation for widely used univariate methods, thereby addressing one of the key practical issues impeding more rigorous evaluation.

The package remains under active development and maintenance and we would like to invite collaboration from the community to extend and further develop its functionalities. Aside from adding further methods, the modularity of the different methods can be further improved, enabling an even more flexible use of different methods by the user. In addition, a systematic review of different available software tools and methods for bias adjustment could be of use to the community. Furthermore, the implications of bias adjustment on the outcomes of impact modelling studies could be examined based on the evaluation

and comparison of different methods within the ibicus package. The ibicus evaluation can also be used as a starting point to

further examine physical sources of climate model biases which can inform improvements in the representation of physical processes within the climate model itself. Also, both the choice of validation period as well as the choice of observational dataset and uncertainty therein have been shown to affect the results of bias adjustment (Casanueva et al., 2020). While this is not explicitly explored in this publication or package, the evaluation tools available through ibicus enable the investigation of these issues.

Finally, the results presented in this paper raise a number of important broader questions regarding the use and future development of bias adjustment methods. The finding that different bias adjustment methods lead to very different results raises the question of whether bias adjustment should be seen as an additional source of uncertainty, as suggested by Lafferty and Sriver (2023). However, the paper also shows that different methods perform better or worse depending on the region and variable studied, which constitutes a clear reason to evaluate and select the bias adjustment targeted to the use case rather than viewing different methods as another source of uncertainty. This then raises questions about whether choosing a 'standard' bias adjustment method to render results comparable is valid and useful in many applications. These questions can serve as a starting point to re-consider both the application of bias adjustment, as well as initiate future development on methods suitable to address the different fundamental issues facing bias adjustment. Existing research avenues include approaches to post-process the entire climate model ensemble (Chandler, 2013; Rougier et al., 2013; Sansom et al., 2021) or conditioning the bias adjustment on specific relevant large-scale processes (Maraun et al., 2017; Verfaillie et al., 2017; Manzanas and Gutiérrez, 2019).

*Code and data availability.* The current version of ibicus is available from PyPI (https://pypi.org/project/ibicus/) under the Apache License Version 2.0, and described in detail under https://ibicus.readthedocs.io/en/latest/. The source code is available via GitHub (https://github.com/ecmwf-projects/ibicus). The exact version of ibicus used to produce the results used in this paper is archived on Zenodo (doi:10.5281/zenodo.8101898, Spuler and Wessel, 2023), as are input data and scripts to run ibicus and produce the plots for all the simulations presented in this paper (doi:10.5281/zenodo.8101842, Wessel and Spuler, 2023). The ERA5 and CMIP6 data used were accessed via the Copernicus Climate Data Store under the Copernicus licence: https://doi.org/10.24381/cds.143582cf and https://doi.org/10.24381/cds.c866074c respectively.

*Author contributions.* JW and FS lead the conceptualization, software development, methodology and formal analysis of the software package and case study, and prepared and wrote the original draft of the paper, contributing equally to all steps outlined. EC and CC lead the project administration and supervision of the project, provided resources and contributed to the writing of the paper by reviewing and editing. JV and EC also contributed to the software development and JV supported the project administration.

*Competing interests.* The authors declare that they have no conflict of interest.

*Acknowledgements.*  We thank the researchers who developed the different bias adjustment methods implemented in this package. In particular, we thank Matthew Switanek for responding to and engaging with our questions during the development period of the package and the discussion during EGU 2023; Matthias Mengel for discussions on the results of the case study. Fiona Spuler and Jakob Wessel thank their PhD supervisors Marlene Kretschmer and Ted Shepherd, and Frank Kwasniok and Chris Ferro, for helpful discussions on the case study; and thank Esperanza Cuartero for her support during the ECMWF Summer of Weather Code programme. We also thank Benjamin Aslan, Simon-

etta Spavieri, Cynthia Rodenkirchen and Philipp Breul for helpful input on the naming of the package. We acknowledge the World Climate Research Programme, which, through its Working Group on Coupled Modelling, coordinated and promoted CMIP6. We thank ECMWF for developing the ERA5 reanalysis. Jakob Wessel is supported by the University of Exeter and funded by the Engineering and Physical Sciences Research Council, Grant/Award Number: 2696930. Fiona Spuler is supported and funded by the University of Reading. The development of this package was funded under the ECMWF Summer of Weather Code 2022 Scholarship for Early-Career Researchers.

**Table A1.** Bias adjustment methods currently implemented in ibicus with variables covered and details on their functioning. Here $x$ refers to observations $x_{obs}$ or climate model values during the historical / reference $x_{cm,\,hist}$ or future period $x_{cm,\,fut}$ and $F$ to a Cumulative Distribution Function (CDF) fitted either parametrically or non-parametrically. Covered variables indicate variables for which the bias adjustment method currently has default settings and climatic variables with a * are variables with experimental default settings. Those are settings that were not published in the peer-reviewed literature but were found to give good performance. The references given are the references used for the implementation of the method in the ibicus package.

| Name | References | Details |
|---|---|---|
| ISIMIP3BASD | Hempel et al. (2013); Lange (2019, 2021a) | **Method**: semi-parametric quantile mapping-based method that aims to be trend-preserving in all quantiles. Generates "pseudo future observations" by applying the models' climate change trend to observations either additively, multiplicatively or in an alternative way. Applies quantile-mapping between the modelled future values and the pseudo future observations, either non-parametrically or parametrically, depending on the variable, optionally with an event likelihood adjustment as in Switanek et al. (2017). The core method is applied in a running window to account for seasonality, and trends in both observations and model are removed prior to applying the method. <br><br> **Covered variables**: hurs, pr, prsnratio, psl, rlds, rsds, sfcWind, tas, tasrange, taskew. |
| CDFt | Michelangeli et al. (2009); Vrac et al. (2012, 2016); Famien et al. (2018) | **Method**: non-parametric quantile mapping that aims to be trend-preserving in all quantiles. CDFt constructs a CDF of future observations and then applies a quantile mapping between the CDF of the future climate model values and the CDF of the future observations: <br><br> $$x_{cm,\,fut} \rightarrow F^{-1}_{obs,\,fut}(F_{cm,\,fut}(x_{cm,\,fut})) = F^{-1}_{cm,\,fut}(F_{cm,\,hist}(F^{-1}_{obs,\,hist}(F_{cm,\,fut}(x_{cm,\,fut})))).$$ <br><br> Because non-parametric CDFs will not be able to map values outside the range of the data an additive or multiplicative shift can be applied to the future and historical climate model data prior to fitting CDFs: the additive or multiplicative bias in the mean can be subtracted / divided out first. CDFt can be run separately for each month of the year to account for seasonality as well as in a running window over the future period, to smooth discontinuities and relax the stationarity assumption. To correct precipitation occurrences in addition to amounts Stochastic Singularity Removal (Vrac et al., 2016) is applied. <br><br> **Covered variables**: hurs*, pr, psl*, rlds*, rsds*, sfcwind*, tas, tasmin, tasmax, tasrange*, taskew*. |
| Scaled Distribution Matching (SDM) | Switanek et al. (2017) | **Method**: parametric quantile mapping that aims to be trend-preserving in all quantiles. Conceptually similar to Quantile Delta Mapping and ECDFM. Scales CDFs by projected absolute (temperature) or relative (precipitation) changes, whilst at the same time also adjusting the likelihood of individual events, by adjusting return intervals, prior to mapping. <br><br> **Covered variables**: pr, tas, tasmin*, tasmax*. |

**Table A2.** Table A1 cont.

| Name | References | Details |
|---|---|---|
| (Detrended) Quantile Mapping (dQM) | Cannon et al. (2015); Maraun (2016) | **Method**: quantile by quantiles mapping of observational and climate model distribution. Forms the basis of most other methods listed. Trends in the mean can be adjusted for using detrended quantile mapping, removing trends before quantile mapping and reapplying them afterwards, either additively or multiplicatively.<br><br>$$x_{\text{cm, fut}} \to F_{\text{obs}}^{-1}\big(F_{\text{cm, hist}}(x_{\text{cm, fut}})\big).$$<br><br>**Covered variables**: hurs*, pr, psl*, rlds*, sfcWind*, tas, tasmin*, tasmax*. |
| Quantile Delta Mapping (QDM) / Equidistant CDF Matching (ECDFM) | Li et al. (2010); Wang and Chen (2014); Cannon et al. (2015) | **Method**: parametric quantile mapping methods that aim to be trend preserving in all quantiles, with special focus on high quantiles. Quantile Delta Mapping applies the following transformation to the future climate model values $x_{\text{cm, fut}}$ if relative changes are to be preserved (eg. for precipitation):<br>$$x_{\text{cm, fut, bc}}(t) = x_{\text{cm, fut}}(t) \cdot \frac{F_{\text{obs}}^{-1}\big(\hat{F}_{\text{cm, fut}}^{(t)}(x_{\text{cm, fut}}(t))\big)}{F_{\text{cm, hist}}^{-1}\big(\hat{F}_{\text{cm, fut}}^{(t)}(x_{\text{cm, fut}})\big)},$$<br>and the following for absolute changes (eg. for temperature):<br>$$x_{\text{cm, fut, bc}}(t) = x_{\text{cm, fut}}(t) + F_{\text{obs}}^{-1}\big(\hat{F}_{\text{cm, fut}}^{(t)}(x_{\text{cm, fut}}(t))\big) - F_{\text{cm, hist}}^{-1}\big(\hat{F}_{\text{cm, fut}}^{(t)}(x_{\text{cm, fut}})\big).$$<br><br>Quantile Delta Mapping for absolute changes is equivalent to the ECDFM method by Li et al. (2010), however the parameters chosen, especially the distributions used for the CDF fits are different. In Quantile Delta Mapping the CDF for future climate model values is fitted in a running window going over the future period to account for long term changes in the trend. Also a running window over the year is included to account for seasonality. This is not the case for ECDFM.<br>**Covered variables**: hurs*, pr, psl*, rlds*, sfcwind*, tas, tasmin*, tasmax*. |
| Linear Scaling (LS) | Maraun (2016) | **Method**: simple correction method adjusting biases in the mean (additive case):<br><br>$$x_{\text{cm, fut}} \to x_{\text{cm, fut}} - \big(\bar{x}_{\text{cm, hist}} - \bar{x}_{\text{obs}}\big),$$<br><br>or mean and variance (multiplicative case):<br><br>$$x_{\text{cm, fut}} \to x_{\text{cm, fut}} \cdot \frac{\bar{x}_{\text{obs}}}{\bar{x}_{\text{cm, hist}}}.$$<br><br>**Covered variables**: hurs*, pr, psl*, rlds*, rsds*, sfcWind*, tas, tasmin, tasmax. |
| Delta Change (DC) | Maraun (2016) | **Method**: technically not a bias adjustment method. Adds a climate model trend to observations either additively or multiplicatively. Similar to Linear Scaling, however it adjusts the observations and not the climate model.<br>**Covered variables**: hurs*, pr, psl*, rlds*, rsds*, sfcWind*, tas, tasmin, tasmax. |

**Table A3.** Treatment of precipitation (pr) dry days of bias adjustment methods currently implemented in ibicus.

| Method | Treatment of dry days |
|---|---|
| ISIMIP3BASD | Explicit adjustment of future dry day frequencies as outlined in Lange (2019) and Lange (2021a). |
| CDFt | Either mapping using the Stochastic Singularity Removal technique (Vrac et al., 2016, default) or using the empirical CDFs. |
| Scaled Distribution Matching (SDM) | Adjustment as in Switanek et al. (2017): Set all values below a certain threshold to zero and explicitly calculate the amount of bias corrected future rainy days. Note: the current method does not support correcting the number of precipitation days upwards, so to transform dry days into wet days. |
| (Detrended) Quantile Mapping (dQM) | Flexible:<br>– Mapping using a censored CDF as in the QDM method.<br>– Mapping using a precipitation hurdle model.<br>– Adjustment of intensities only. |
| Quantile Delta Mapping (QDM) | Adjustment as in Cannon et al. (2015): 1) Randomize values between 0 a a fixed threshold, 2) Fit censored parametric CDFs assuming values below the fixed threshold are censored, 3) Apply the QDM method using the CDFs and set values under the threshold to zero again. |
| Equidistant CDF Matching (ECDFM) | Flexible:<br>– Mapping using a censored CDF as in the QDM method.<br>– Mapping using a precipitation hurdle model.<br>– Adjustment of intensities only. |
| Linear Scaling (LS) | Currently no explicit adjustment of dry days. |
| Delta Change (DC) | Currently no explicit adjustment of dry days. The number of dry days stays the same as in the observations. |

**Table B1.** Overview of CMIP6 models and their model developers used in the case study in section 4.

| Model Name | Institution |
|---|---|
| ACCESS-CM2 | Commonwealth Scientific and Industrial Research Organisation / Australia |
| CMCC-ESM2 | Euro-Mediterranean Centre on Climate Change / Italy |
| IPSL-CM6A-LR | Institut Pierre-Simon Laplace / France |
| MIROC6 | Japan Agency for Marine-Earth Science, University of Tokyo, National Institute for Environmental and RIKEN Centre for Computational Science / Japan |
| MPI-ESM1-2-LR | Max Planck Institute for Meteorology / Germany |
| MRI-ESM2-0 | Meteorological Research Institute / Japan |
| NORESM2-MM | Norwegian Climate Centre / Norway |

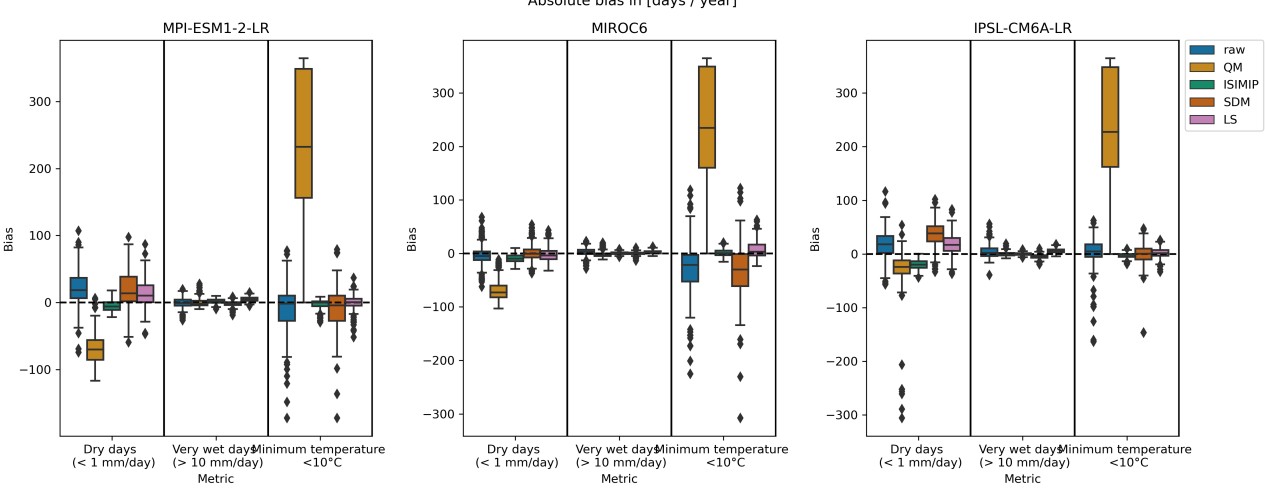

**Figure A1.** As figure 2, but including Quantile Mapping for minimum temperature.

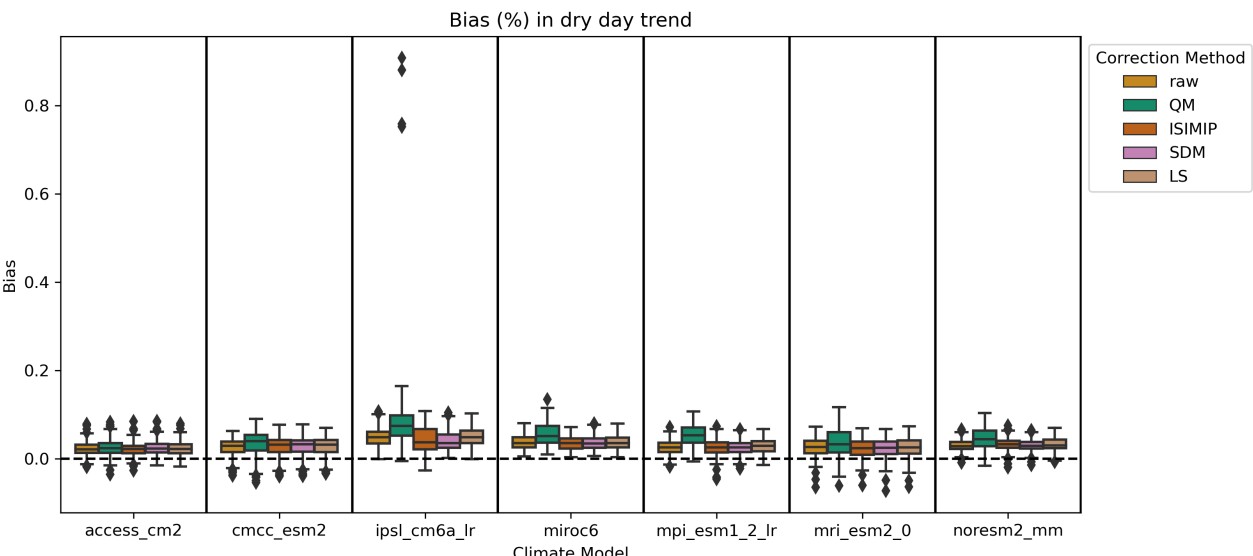

**Figure A2.** As figure 6, but without axis limits at +-100.

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
