# Peer review of "ibicus: a new open-source Python package and comprehensive interface for statistical bias adjustment and evaluation in climate modelling (v1.0.1)"

_EGUsphere, 2023_

## Author Comment (AC1)

**Reviewer #1 - 'Comment on egusphere-2023-1481' - September 14th, 2023**

*Comments by the reviewer in blue italics* - responses by the authors in black

Dear reviewer,

Thank you for your constructive criticism of the GMD paper submission and the software package. We have addressed your comments, please find details of our response below.

*I find the effort of making this tool for evaluating and inter-comparing bias correction methods important and highly relevant. The paper is well structured, and although I haven't tested the software itself, it seems from the manuscript that it is producing many relevant statistics and useful plots. The manuscript lacks important details, and I have several suggestions for how to compare the bias correction methods in my major comments, which I wish the authors to consider and make a revision.*

*Major comments:*

*I think the idea of implementing multiple methods in one software is a good idea, as is the common framework for evaluation.*

*One major remark I have is that some components are not tied to a certain method. One example is the treatment of dry days, where the threshold and the way they are bias corrected will have impacts on certain metrics that have nothing to do with the method applied to the rest of the distribution. An example is the results for your "QM" method. I suggest that this component is detached in a way that the same treatment is applied to all methods when making the inter-comparison. The dry day corrections themselves can be assessed separately. Another example is how extremes and data outside the calibration data range are handled. This is often not properly defined for different methods, but can have large consequences for indicators based on the extreme ends. When possible, the same tail handling should be applied to all (empirical) methods. If possible to implement in your software, this would make it a very useful tool to assess and find the method best suited for a particular case of bias correction. I am not expecting you to implement this for a new revision, but please think about it and add to a discussion section on future developments.*

Response: We thank the reviewer for this helpful comment. We agree with the comment, and the modularity of different methods was certainly a key challenge we discussed in detail when we were developing the architecture of the ibicus software.

In principle, we agree that many of these different choices are interchangeable across methods (within certain limits). However, practically speaking and after some deliberation when developing the package, we decided that it would not be feasible to make the package fully modular (to enable, for example, the application of the same dry day treatment or handling of extremes across methods) for two related reasons: one is that different choices can be entangled, and the second is that some choices such as the dry day treatment can be seen as 'core characteristics' of a specific method. For example, in the case of the ISIMIP method, the dry day treatment fits into their general approach of bias adjusting bounded variables and is being developed further in new versions of the method (see

Lange 2021 eq. 1 in comparison to Lange 2019 eqs. 8 and 9). In the case of Scaled Distribution Mapping (SDM), the dry day treatment is defined as part of the core characteristics of the method itself (the number of rainy days is mapped using formula 1 defined in Switanek et al. 2017) and in the case of CDFt, the Stochastic Singularity Removal technique corrects both occurrence and intensity jointly non-parametrically based on Vrac et al. (2016).

The approach we therefore took in the design of the ibicus software architecture design was to define some components as core characteristics of each method, making selected options interchangeable for specific methods but not for others. This means the package ends up being partially modular. For example, the dry day treatment can be modified in the Quantile Mapping method: options range from a mapping using a censoring approach similar to the one outlined in Cannon et al. 2015, a fit of a precipitation hurdle model together with randomization to be able to map dry days into wet ones and an adjustment of intensity only as well as possible user provided different adjustments.

We agree with the reviewer that it would be a good idea to develop the modularity of the different methods further in future versions of the ibicus software, and we also agree that the discussion of the modularity of the different methods was previously not detailed enough in the paper. We have added several sections in the text (in the background section and description of the ibicus package) that highlight the modularity of methods, using the treatment of dry days as an example. Furthermore, we expanded table 1 that gives an overview of the choices made in different methods to include the treatment of dry days and extremes and added an additional table in the appendix to detail the treatment of dry days in different methods. For each individual method (family), the package documentation gives a further detailed overview of adjustments that are possible within the ibicus package.

Regarding the case study presented in the paper, our aim was to compare different methods as they are commonly applied. For ISIMIP and SDM, the dry day treatment is, in our interpretation, part of what defines the method in how it is most commonly applied, and we therefore chose to not change these aspects of the method in the case study. For QM and LS we have added some text commenting on the dry day treatment used.

Text modifications:

- Background and ibicus description see latex diff document
- New row in table 1: "Treatment of dry days and extremes – Methods have different ways of handling certain aspects of the distribution such as extreme values or dry days in the case of precipitation. For extremes some methods use an extrapolation based on parametric distribution, which can be modified by the user for example should a mapping based on extreme value theory be required. For dry days the ISIMIP, SDM and CDFt methods provide an explicit handling that might be appropriate in some situations but not in others. QDM treats the mapping of dry days as a censoring problem and adjusts them together with the body of the distribution whilst for methods like QM and ECDFM the user has the choice of different treatment methods."
- New table on the treatment of dry days in the appendix.

*I would like if the authors can add some information on how a user can implement their own method in Ibicus, what are the steps? Is there a guide in the software documentation etc. And why is It called ibicus?*

Response: We thank the reviewer for this idea and agree that a guide for users on how to implement a new method in ibicus would be very useful to have in the software documentation. While such a guide is outside the scope of this GMD paper, we will include a new tutorial in the software documentation with the next ibicus release.

The package is called ibicus because we felt that it was a suitable name for a software package. It is not an acronym but rather just sounds similar to abacus as well as ibis which is the name of a bird and therefore in line with package naming conventions at ECMWF.

*The method called "Quantile Mapping" is not properly defined and named. This is a category of methods that includes most of the methods that are used in this paper. A more precise name is necessary, and also a detailed description of how the quantile mapping is implemented, e.g. which quantiles, how extremes are dealt with, and especially for data outside of the calibration range.*

Response: We agree that a large number of choices are possible when applying QM leading to fairly different "flavours" of the method and a "family" of QM methods (as detailed in the comment above regarding the modularity of different methods). In fact, most methods in the package have some sort of quantile mapping in this form as their core which is then expanded in various ways. Related to the idea of making the package partially modular (see response above), we believe that it is useful to include a generic quantile mapping method that can be modified in many different ways.

Based on publications such as Maraun (2016), we define Quantile Mapping (QM) as a fairly concrete method based on the mapping of two cumulative distribution functions as $F_{cm,fut}^{-1}(F_{cm,hist}(obs))$, as detailed in the appendix table 1, as well as the documentation of the software package. We believe that despite the many different variations of Quantile Mapping the package offers, this defines it as a method. We agree with the reviewer that further specification is necessary when applying the method in the case study and have added an additional sentence in the case study providing details of the type of quantile mapping applied there.

*Generally, the authors need to more clearly defined how each method deals with dry days as this can have a large impact on some of the statistics. Please revise with some statements about this in the main text and in Table A1.*

Response: We have adjusted the main text in a couple of locations (background section, ibicus description section and table 1, see below). As the table A1 is supposed to outline the implemented methods for all 10 supported climatic variables we opted against including additional information on the treatment of dry days directly in there as this would impede readability. Instead, we added a second table A2 containing additional information about dry day treatments. We also adapted the text in section 4 (the case study) stating what dry day treatment was used for the individual methods. As for ISIMIP and SDM the dry day treatment is closely entangled with the method design; a conscious choice was only made for QM, which we state in section 4.1:

New text in italic: "These four methods are applied to daily total precipitation (pr) and daily minimum near-surface air temperature (tasmin), chosen to cover two different types of variables (bounded vs unbounded, different distributions etc) that are both highly relevant for many impact studies. *The bias adjustment methods are used with their ibicus default settings for both variables (for more details see table A1 and the software documentation). This means that the ISIMIP and SDM methods provide an explicit adjustment of dry day frequencies, whilst for QM they are treated as censored and the method based on Cannon et al. (2015) is applied and LS provides no explicit adjustment, scaling all values.*"

Finally, we have added some words on future developments in the discussion section highlighting that extending the modular aspect of certain methods and the customizability is something that we plan to implement in future versions of ibicus and that will increase comparability of bias adjustment methods.

*It is not clear what future is for which the climate trends are calculated. Is all done between periods in the historical range 1959 to 2005? I cannot find any other information about time periods, nor any information about SSP-scenarios used. Please clarify this point.*

Response: We have adjusted the text as follows:

"Table B1 in the appendix provides more details on these models. We used the historical runs as well as the SSP5-8.5 experiments."

Original text: The data ranges from January 1st, 1959 to December 31st, 2005, with the initial 30-year period (1959-1989) serving as the historical/reference period and used as a training dataset and the subsequent 15-year period (1990-2005) used for validation purposes.

Modified text: "The historical data ranges from January 1st, 1959, to December 31st, 2005, with the data from January 1st, 1959, to December 31st, 1989, serving as the historical/reference period and used as a training dataset and the subsequent period: January 1st 1990 to December 31st 2005 used for validation purposes. Bias adjustment is applied to the validation period as well as the future period: January 1st, 2080, to December 31st, 2100."

*It is also important to state something about the magnitude of the climate trend, as it gives some information about the signal to noise levels and whether differences between methods are significant or not.*

Response: Thank you for this comment, the magnitude of the climate change trend is indeed relevant to specify when showing these results. We have added the range of the raw trend across locations for both dry days and mean minimum daily temperature in the respective figure captions.

Added sentences in the figure captions:

Figure 6: [...] The magnitude of the raw projected change in dry days depends on the climate model and, across different locations, lies between 10 fewer and 30 more dry days on average per year.

Figure 7: [...] The magnitude of the raw projected change in mean minimum daily temperature again depends on the climate model and, across different locations, lies between 2-5K.

*Detailed comments:*

*L22: "an empirical transfer function" this could equally well be a parametric, so please remove the word "empirical".*

Response: Thank you for your comment, we changed the term empirical to statistical in all instances. We had originally called the transfer function empirical as it is 'based on data' as opposed to 'based on theory', even if a parametric fit is used. We had wanted to avoid using the term statistical as it sometimes implicitly implies machine learning methods as an alternative. However, based on the comments of both reviewers, we concluded that using the term empirical here is potentially misleading and therefore changed it.

*L26: This sentence is a bit difficult, which has to do with the vague part "ranging from" which lists generic reports which are not throughout using bias adjustment. Please reformulate and be more precise in your statement.*

Response: We disentangled the sentence as follows:

Original text: Despite widespread use, ranging from the IPCC AR6 WGI&II (IPCC, 2021, 2022) reports to the climate scenarios used by central banks across the world (NGFS, 2021), Maraun et al. (2017) and others highlight fundamental issues with statistical bias adjustment and show that the approach can destroy the spatiotemporal and inter-variable consistency of the climate model and is prone to misuse.

New text: Despite widespread use both within the scientific community (see, for example, IPCC, 2021, 2022), as well as by climate service providers and practitioners (see, for example, climate scenarios used by central banks across the world, NGFS 2021), bias adjustment is known to suffer from fundamental issues. These issues have been highlighted, among others, by Maraun et al. (2017) who show that bias adjustment not only has limited potential to correct misrepresented physical processes in the climate model but can also introduce new artefacts and destroy the spatiotemporal and inter-variable consistency of the climate model.

*L61: please remove "empirical".*

Response: Corrected, see comment above.

*L114: "MIdAS" according to the reference.*

Response: Corrected in the text.

*L150: Please clarify what is meant with "optional data information for running windows".*

Response: We have adjusted the sentence as follows:

"This takes a 3-dimensional numpy array of observations, as well as historical and future climate model simulations as input. Bias adjustment methods using a running window might require date information as 1-dimensional arrays."

*L153: In which contexts are these methods "most widely used"? Can you quantify this?*

Response: We have adjusted the sentence (see below). Furthermore, we have included a sentence in the future developments section that ibicus might be extended for more methods in future releases.

New text: "The methods were chosen to cover some of the most widely used bias adjustment methods. These methods were cited thousands of times, have been used in major coordinated projects like the ISIMIP project (Lange, 2019) and are based on fairly different assumptions, making them suitable for different applications.

*L157: "Quantile Mapping". I do not think this is a good way to describe this method in contrast to the others. They are all of the quantile mapping family, and there is no single clear definition of what quantile mapping is, but it needs to be clearly defined. If you are referring to detrended quantile mapping (as in Table A1), you could use the abbreviation DQM instead.*

Response: Please see the response on the comment above. To differentiate plain Quantile Mapping from the mean trend preserving variant (detrended Quantile Mapping – dQM) we prefer to use the abbreviation QM here.

*L170: The case is a bit more complicated when threshold-based indicators are used. Then it is not possible, or wanted, to preserve the original trend.*

*&*

*L268: Note no single method is attempting to preserve trends for all possible indicators, but target a single or more moments or quantiles of a distribution.*

Response: We completely agree with these comments. Regarding the first comment, we added a more detailed discussion of trend modification through bias adjustment in the background section. Regarding the second comment, we argue not that we expect the trend in all indicators to be preserved, but rather that it is necessary to evaluate how the trend is modified. By pointing out that even trend-preserving methods can change climate change trends we want to convey the two points highlighted by the reviewer, namely: 1) No method will preserve the trend in every indicator. In our experience, this is quite commonly misunderstood by users of trend-preserving bias adjustment methods. For example, the preservation of the trend in dry days can be stated in the publication of some methods, and assumed when the method is used, but not evaluated (eg. Lange, 2019). 2) Even if a method aims to preserve certain trends the success of this is dependent on whether the underlying assumptions of the method are met.

*L195: It is not clear what time periods are used, and what the "future" is for which the trends are assessed. See major comment.*

We have added the following:

"Table B1 in the appendix provides more details on these models. We used the historical runs as well as the SSP5-8.5 experiments."

"The historical data ranges from January 1st, 1959 to December 31st, 2005, with the data from January 1st 1959 to December 31st 1989 serving as the historical/reference period and used as a training dataset and the subsequent period: January 1st 1990 to December 31st 2005 used for validation purposes. Bias adjustment is applied to the validation period as well as the future period: January 1st 2080 to December 31st 2100."

*L209 and 210: There are 31 and 16 years in these periods. Please check your statements.*

Response: That is correct, we have amended the text accordingly (see response to the last comment).

*L215: Please define "temperature" in this sentence. Is tasmin still indended, or some other temperature measure?*

Response: We have adjusted the sentence to refer specifically to tasmin:

"– tasmin greater than the seasonal 95th percentile of the daily minimum temperature in each grid cell during the historical period (1959- 1989). This can be an indicator of the impacts of heatwaves (Raei et al., 2018)."

*L222: Bias should be near zero for the calibration period, and it would be good to know if that is the case as it is a confirmation that the implementation is correct.*

Response: We tested the implemented methods extensively and, for example, provide a notebook showing the correspondence of our implementation of the ISIMIP method with the reference implementation. However, it is important to note that the bias on the calibration period is not necessarily zero if the method is implemented correctly: for example, if the parametric fit does not fit the data well or other method-specific assumptions are not met, a correct implementation of the method might result in "residual" or in extreme cases even increased bias over the calibration period.

*L232 and Figure 2 – dry days. It is necessary to explain how dry days are handled in the different methods to understand what is happening with "QM".*

Response: We have included some text outlining the dry day treatment of the different methods in the case study (see above).

*L237: I do not understand the use of the word "assimilate" here. Please reformulate of explain.*

Response: We changed the word, adjusting the sentence as follows:

"When investigating the spatial distribution of the bias (figure 3), we find that certain methods can homogenize the spatial pattern of the bias across climate models."

*Figure 3: This plot would be more efficient with model names only at the left and method names only on the top, and larger panels. If it is a direct output of the software, you can state that as it will justify the less optimal layout.*

Response: The plot itself is not a direct output of the software as by operating on a numerical level it does not have information on the geographical position. We have tried adjusting the figure with model names on the left and method names on top, however that actually decreases the possible size of each individual panel. For legibility reasons we therefore opted to keep the current layout.

*L260: Please defined the time periods used and if any emission scenario was used. Some measure of signal to noise or significant would be good to include as well.*

Response: we added text on the time periods and emission scenarios (see responses above), and made amendments regarding the trend signal to noise question, detailed in the response to the comment above.

*L275: Again, please defined the dry day definitions and treatment for each method as it has large impacts on the results, and shall in my opinion not be confused with the general method for the rest of the distribution.*

Response: Our response and amendments to the text are covered by our response to the related comments above, as well modifications already made to the text on the basis of these other comments.

*Figure6-caption: "change in the number of dry days" right?*

Response: We changed the text, it now reads:

"Distribution of location-wise change in the additive climate trend in dry days introduced through the bias adjustment method, computed by computing the additive trend between the validation period and the future period in both the raw and the bias adjusted model and taking the percentage difference between the two trends."

*Table A1: last sentence in CDFt "[SSR] can be applied." But is it applied here?*

Response: This is applied as default setting; however, the user can choose to deactivate it in which case the full distributions (including dry days) are mapped using the CDFt method. We have changed the sentence to:

"To correct precipitation occurrences in addition to amounts Stochastic Singularity Removal (Vrac et al., 2016) is applied."

---

## Author Comment (AC2)

**Reviewer #2 - 'Comment on egusphere-2023-1481' - October 2nd, 2023**

*Comments by the reviewer in blue italics* - responses by the authors in black

Dear Jorn Van de Velde,

Thank you for your praise as well constructive criticism of the GMD paper submission and the software package. We have addressed your comments, please find details of our response below.

*First, I would like to say that I'm impressed by the paper. Further professionalization and evaluation of bias adjustment is clearly necessary, and the authors make an important step forward by providing this software package. In general, the paper is clearly written and provides good examples and results of the code. However, there are still some major and minor remarks that I would like to see discussed and implemented in the paper.*

*General comments*

*Implications of software like this. Further standardizing (or at least standardizing evaluation) becomes clearly possible through this method. This has some consequences. First, it allows for answering questions on the seemingly 'detailed' components of methods, such as the applied time windows (e.g. seasonally vs. 90 days vs. 60 days), number of years for calibration and evaluation, number of data points selected. When implementing a new method, these questions are often sidelined, but they could affect the final result. Not to say that they do, but at least it should be evaluated through standardized tools.*

Response: Thank you for this point. In response to both reviewers, we added text in the background section and ibicus description that discusses the modularity of different methods (see also our next comment). We also amended the discussion section in response to both your and Richard Chandler's comments, and, amongst other things, mention that ibicus can be used to explore the consequences of these different choices that might impact the results.

*Second, building on one of the comments of Anonymous Referee #1, some method components are not tied to a certain method. This might be considered to be a slightly more philosophical note, but it is possible to consider a switch from methods as 'packages' to methods as 'build from a set of elements'. As elements, I consider e.g. the choice of distribution(s), the choice of dry-day treatment, the order in which steps are taken, additional post-processing steps… Software like this may thus eventually help to disentangle methods and compare their elements (and changes to these elements). Even if they were not originally implemented as such (e.g. a distribution not foreseen by the original author, or an additional post-processing step applied in another software package). According to the documentation, it seems that the way the code is set up, allows (to some extent) for this kind of experimentation.*

Response: We thank both reviewers for the constructive comments regarding the modularity of methods. As we also mention in our responses to reviewer #1, this was a question we deliberated on while developing the package. We agree that many components are not tied to a specific method and as you note, we tried to reflect that in the design of ibicus by giving the user options to modify components such as the dry day treatment for some of the methods. However, as we note in the

response to reviewer #1, we also retained different core characteristics for the different methods (for example we consider the dry day treatment a core characteristic of some methods such as SDM) to ensure the recognisability of some methods such as ISIMIP, and because some of the choices are entangled with each other which stands in the way of making the package fully modular. In response to the reviewers' comments, we added additional text on the modularity of methods in the background section and description of the ibicus package, extended table 1 to include a high-level overview of dry day treatment, and added an additional table in the appendix that details the treatment of dry days for the individual methods. However, while we attempted to provide a detailed description of all the different modifiable components in the software documentation, we believe that such a detailed description would be beyond the scope of this GMD submission. Based on your comments, we plan to further improve the modularity of the package in future releases and mention this in our amended discussion section.

*To conclude, software like this could in time change and influence how we evaluate bias adjustment methods. Could you comment on this and discuss this in your paper? That would certainly further enhance the discussion/conclusions of this paper. Or would even merit a separate discussions section, as Richard Chandler also touches upon this point in comment #3.*

Response: We thank all reviewers for highlighting the potential implications of this package as well as possibilities for future methodological development. We significantly amended the discussion section to discuss the implications of the package.

*To take the previous point even one step further, it would be relevant to actually review and compare existing software packages. This is seriously out of scope for this paper, but it might be relevant mentioning this need in the discussion/conclusion.*

Response: We agree and added a sentence on this in the new discussion section in the paragraph on future development.

*Although the authors have taken the time to get acquainted with some of the important discussions in bias adjustment/statistical downscaling literature and touch upon a lot of subjects, I think there is still a lot of ground left to cover. If a reader interested in applying bias adjustment software starts from your paper, it should be possible to track down most of the papers discussing issues and steps forward. So far this is not always possible. In the specific comments, I have given some references related to topics discussed at specific points, which I think are all relevant to refer to in the paper.*

Response: We thank Jorn Van de Velde for highlighting some additional literature that is relevant to refer to and his concrete suggestions. We have included a large number of these (see details below).

*Additionally to reading the paper, I also did a check of the documentation and tutorials. It seems like a lot of work went into this, for which I would like to congratulate you. Given the amount of information available, I hope a lot of potential users and contributors will find, apply and contribute to your package! However, take note of the changes and additional literature suggested for the paper, and also implement them in the documentation.*

Response: Thank you very much for the appreciation of the work that went into the documentation. We will include any modifications made in the paper in the documentation of the package and will also link this paper in the software documentation as background material for users.

*Note that 1) I agree with most of the comments of the other reviews (so far posted) and would like to see them addressed properly. Only where really necessary, I repeated a comment. 2) I consider this to be minor revisions, as the software, evaluation set-up and main conclusions are coherent and scientifically sound, but reading the suggested papers might of course take some time.*

*Detailed comments*

*L19: it might be good to provide a few examples for the interested reader. See e.g. Vautard et al. (2021) or Galmarini et al. (2019) for relatively recent papers discussing respectively model biases and the impact on agriculture.*

Response: We have expanded this point a bit and included a few more examples:

Original text: "Even though climate models have greatly improved in recent decades, simulations of present-day climate still exhibit biases. This means that there are systematic discrepancies between model output and observations that become especially relevant when using the output of climate models for local impact studies, for example by running agricultural or hydrological models."

Modified text: "Even though climate models have greatly improved in recent decades, simulations of present-day climate of both global and regional climate models still exhibit biases (Vautard et al. 2021). This means that there are systematic discrepancies between statistics of the model output and observational distribution (Maraun, 2016). These discrepancies in the two distributions become especially relevant when using the output of climate models for local impact studies that often require focus on specific threshold metrics such as dry days, for example when running hydrological (Hagemann et al. 2011) or crop models (Galmarini et al 2019)."

*L22: I would like to stress the comment by AR#1. There are many examples of parametric transfer functions out there.*

Response: We have changed empirical to statistical in all instances. We had originally called the transfer function empirical as it is 'based on data' as opposed to 'based on theory', even if a parametric fit is used, but realised based on comments of both reviewers that using the term empirical here is potentially misleading. We had originally wanted to avoid using the term statistical as it sometimes implicitly implies machine learning methods as an alternative.

*L24: many multivariate methods as well build on quantile mapping (e.g. by first applying univariate quantile mapping and then a multivariate adjustment procedure, the so-called marginal/dependence multivariate bias adjustment)*

Response: This is true, and we have slightly modified the sentence. However, as this is still the introduction, we mostly want to focus on highlighting the breadth of methods, without going into too much detail yet.

Original text: "A variety of statistical bias adjustment methods have been developed and published in recent years, ranging from simple adjustments to the mean, to trend-preserving adjustments by quantile and multivariate methods (Michelangeli et al., 2009; Li et al., 2010; Cannon et al., 25 2015; Vrac and Friederichs, 2015; Maraun, 2016; Switanek et al., 2017; Lange, 2019, and more)."

Modified text: "A variety of statistical bias adjustment methods have been developed and published in recent years, ranging from simple adjustments to the mean, to trend-preserving adjustments by quantile and further multivariate adjustments (Michelangeli et al., 2009; Li et al., 2010; Cannon et al., 25 2015; Vrac and Friederichs, 2015; Maraun, 2016; Switanek et al., 2017; Lange, 2019, and more). "

*L25: with regards to multivariate methods, I had to wait until L304 and further to find clarity on why multivariate methods where not implemented here. Although I understand the choice, it should be clear from the start, given the importance of multivariate methods (e.g. in relation to compound events).*

Response: We have included the following sentence in the introduction after the sentence modified in the last reviewer comment. However, we kept a detailed discussion in the original position in the paper.

"While this paper focuses primarily on methods that are applied at each grid cell individually, the use of multivariate methods is further discussed in section 5."

*L40 and further: I could nowhere find a clarity on the implementation of the bias adjustment methods. Did you copy-paste them from existing code, implement them yourselves, or mix them? Did you compare results with the original code (whenever available) or contact the original authors to check the original code? Given that small differences in code implementation can have a potentially large impact, this has to be clear from the start (especially in a journal like GMD)*

Response: The bias adjustment methods were implemented by the authors using the literature describing the individual methods and possibly available reference implementations in R or Python. Choices were made by the authors of ibicus primarily regarding the question of which aspects of the method to modularize. In case of divergence between the literature description and available reference implementations, the respective authors of the method in question were contacted and after the first alpha release of the software package, all bias adjustment developers were invited to comment on the implementation and review the package. Finally, extensive testing was done to ensure the correctness of outputs and consistency of implementations. We have amended the text in two locations to highlight this:

Introduction:

Original text: "The contribution of ibicus is two-fold: It provides a unique unified interface to apply eight different peer-reviewed and widely used bias adjustment methodologies, including Scaled Distribution Matching (Switanek et al., 2017), CDFt (Michelangeli et al., 2009) and ISIMIP3BASD (Lange, 2019)."

Modified text: "The contribution of ibicus is two-fold: For one, it introduces a unique unified interface to apply eight different peer-reviewed and widely used bias adjustment methodologies. The implemented methods include Scaled Distribution Matching (Switanek et al., 2017), CDFt (Michelangeli et al., 2009), Quantile Delta Mapping (Cannon et al. 2015) and ISIMIP3BASD (Lange, 2019)."

Start of section 3:

Original text: "ibicus implements eight state-of-the-art, peer-reviewed bias adjustment methodologies in a common interface that enables the user to modify aspects of an individual methodology to suit their target variable, region and impact of interest."

Modified text: "ibicus introduces a unified, modular, software architecture within which eight state-of-the-art peer-reviewed and widely used bias adjustment methodologies are implemented. This enables researchers to apply different methods through a common interface, and modify components of the methods, such as the treatment of dry days, based on region and impact of interest. The code implementation of each methodology is based on the cited academic publication, as well as available accompanying code that was re-written and modularised to fit the developed interface. Consistency with the original implementation was ensured through rigorous testing and correspondence with the authors of the different methodologies."

*L45: Did you consult Maraun et al. (2015) on the aspect of evaluation and the validation tree? They build heavily on the dimensions you mention here, and follow this up in all papers of the VALUE experiment (see e.g. Maraun et al. (2019)). Although this experiment focuses more heavily on statistical downscaling instead of bias adjustment, the latter is also accounted for and the general principles and lessons should at least be mentioned in a paper on bias adjustment evaluation.*

Response: We have included a reference to the VALUE experiment when introducing the evaluation framework in section 3.3:

Original text: "The ibicus evaluation framework offers a collection of tools to identify these issues and compare the performance of different bias adjustment methods for variables of interest."

Modified text: "The ibicus evaluation framework offers a collection of tools to identify these issues and compare the performance of different bias adjustment methods for variables of interest, building on previous efforts such as the VALUE evaluation framework for statistical downscaling (Maraun et al. 2019)."

*L50: Here, you apply the standard 'section' titles, whereas further in the paper, you refer to sections as 'chapters'. I prefer the former, as it is more standard.*

Response: Thank you for your comment, we changed chapter to section everywhere in the text.

*L69: delta change is not limited to linear scaling. It is more correct to consider delta change as a principle of philosophy, where, in contrast to bias adjustment, not the climate model output is adjusted, but historical time series are adjusted. See e.g. Olsson et al. (2009) or Willems and Vrac (2011) for papers building on this principle.*

Response: This is a helpful point. We included notes on the modularity of different methods and the fact that they are method families rather than methods at several locations in the text (see response to other comments). With regards to the delta change approach, we have adjusted the text to the following:

Original text: "The most common approaches to the bias adjustment of climate models include a simple adjustment of the mean (Linear Scaling or Delta Change), a mapping of the two entire cumulative distribution functions (Quantile Mapping), or more advanced methods that also aim to preserve the trend projected in the climate model (such as CDFt or ISIMIP3BASD). The practice of using bias adjustment methods to also downscale the climate model has been criticized in various publications (von Storch, 1999; Maraun, 2013; Switanek et al., 2022), therefore this paper focuses on bias adjustment of climate models purely for the purpose of reducing biases at constant resolution."

Modified text: "The most common approaches to the bias adjustment of climate models include a simple adjustment of the mean (Linear Scaling), a mapping of the two entire cumulative distribution functions (Quantile Mapping), or more advanced methods that also aim to preserve the trend projected in the climate model (such as CDFt or ISIMIP3BASD). An alternative approach, often termed Delta Change method, adjusts the historical observations to incorporate the climate model trend (see, for example, Maraun et al. 2016, Olsson et al. 2009 or Willems and Vrac 2011). The practice of using bias adjustment methods to also downscale the climate model has been criticised in various publications (von Storch, 1999; Maraun, 2013; Switanek et al., 2022), therefore this paper focuses on bias adjustment of climate models purely for the purpose of reducing biases at constant resolution."

*L90: given the relative importance of trend preservation in your paper and evaluation, I think this concept should be discussed more in-depth. Consider for example Ivanov et al. (2018), which do not entirely seem to agree with Maraun (2016) (which you refer to), Hagemann et al. (2011) or Casanueva et al. (2018).*

Response: We thank you for pointing out some limitations of our discussion of trend preservation which we agree should be discussed more in-depth. We have expanded the paragraph in the background section significantly, modified Table 1 slightly and included a number of additional references:

[revised manuscript text omitted]

*Table 1: the literature concerning the bias stationarity assumption has been growing recently. In the context of evaluation, some of these papers should be referred to explicitly. Consider e.g. Dekens et*

*al. (2017), Christensen et al. (2008), Chen et al. (2015), Hui et al. (2019), Chen et al. (2020), Wang et al. (2018), Van de Velde et al. (2022) and references therein.*

Response: We agree and have added citations to Van de Velde et al. (2022), Maurer et al. (2013), Chen et al. (2015) and Hui et al. (2015) (see response above).

*L288: There is a very relevant discussion on the issue of uncertainty in Maraun and Widmann (2018). I think it would be a proper addition to your paper.*

Response: Thank you for this comment. We agree that there were points missing from this paragraph discussing uncertainty and have amended the paragraph as follows:

Original text: Figure 8 shows that the climate model ensemble spread of the trend in mean seasonal precipitation is modified when applying bias adjustment. This means that not only the trend but also the range of uncertainty and possible worst-case scenarios analysed in impact studies depend on the bias adjustment method used to pre-process the climate model. As shown in the previous sections, the 'best' bias adjustment method for a given use case depends on the variable, region and impact variable studied. The result shown in figure 8 demonstrates that bias adjustment can add an additional source of uncertainty if the method is applied blindly and not evaluated properly. Interestingly the uncertainty range is not necessarily narrowed as has been postulated by some authors (Ehret et al., 2012), but even extended and shifted in some cases.

Modified text: Figure 8 shows that the climate model ensemble spread of the trend of mean seasonal precipitation is modified in different ways by different bias adjustment methods which is in line with previous findings in the literature (Maraun and Widman 2018, Lafferty et al. 2023). Interestingly the variation (often interpreted as the uncertainty range) is not necessarily narrowed as has been postulated by some authors (Ehret et al., 2012), but even extended and shifted in some cases. From this finding, it follows that the range of uncertainty and possible worst-case scenarios analysed in subsequent impact studies might depend on the bias adjustment method used to pre-process the climate model.

The interpretation of this shift in uncertainty is related to the previously discussed questions on trend preservation, namely whether the change in the climate model trend through a statistical bias adjustment method is justified or not. This issue was mentioned by Maraun and Widmann (2018), who discuss that a minimum requirement to justify a change in the uncertainty spread through bias adjustment should be a critical evaluation of the validity of the results and the assumptions of the underlying statistical model. Given the finding in the previous section, namely that the best bias adjustment method depends on the variable, region and impact variable studied, it follows that indiscriminately applying a bias adjustment method across regions and variables without evaluation can shift the spread of the results of subsequent impact studies in a non-justified manner.

Added citation:

- Lafferty, D.C., Sriver, R.L. Downscaling and bias-correction contribute considerable uncertainty to local climate projections in CMIP6. *npj Clim Atmos Sci* **6**, 158 (2023). https://doi.org/10.1038/s41612-023-00486-0

*L311: François et al. (2020) (which you refer to earlier in the paragraph) should also be referenced w.r.t. the difficulties with multivariate methods, as could Van de Velde et al. (2022)*

Response: Thank you for mentioning these additional references, we added them to the modified discussion section in the amended document.

*Table A1: 1) How were the experimental settings found and defined? Could you give a more expanded explanation? 2) are the references considered to be 'the' references, or just 'standard' references. Especially for linear scaling and delta change (but also quantile mapping), much older references are also available, but are also potentially less clear on the implementation. Please clarify this. 3) Wang and Chen (2014) also further expand on ECDFM and provided the first implementation of the relative version. 4) Cannon et al. (2015) discuss that ECDFM and QDM are practically equivalent. Is this also clear from your evaluation? If not, how come? 5) QDM is at the moment one the commonly applied quantile mapping methods (especially in multivariate methods, see e.g. Mehrotra and Sharma (2016), Nguyen et al. (2016), Cannon (2018)). This could be discussed in function of your evaluation.*

Response: 1) The experimental settings were defined based on variable characteristics (bounded, etc), the default settings implemented for these variables in other methods, and manual testing by the package authors. 2) The references indicated are the ones used for the implementation of the method in the ibicus package. We changed the text in the caption of table A1 to specify this. 3) We have added a reference to Wang and Chen (2014) to the references under ECDFM. 4) The "core" mapping of the two methods, QDM and ECDFM are equivalent which is why we describe them jointly in Table A1. We also find that the two methods produce similar results. However, the results are not equal as the two methods use different types of distributions for the parametric CDF fits and QDM in contrast to ECDFM includes a running window over the future period. 5) This was our reasoning to include QDM, and we mention in the main text that we implemented some of the most widely used bias adjustment methods. We did not add an extra comment in the Table.

New caption text: The references given are the references used for the implementation of the method in the ibicus package.

*References: please clean up your reference section. There are too many 'book:' and 'publisher:' in there, unless this is the current style adopted by GDM*

Response: Thank you for pointing this out. We have improved our reference section.

---

## Author Comment (AC3)

**Richard Chandler - 'Comment on egusphere-2023-1481' - October 2nd, 2023**

Comments by the reviewer in blue italics - responses by the authors in black

We thank Richard Chandler for his comments which have strongly improved the manuscript. We provide detailed responses below.

*This paper makes a welcome contribution to the sometimes murky world of statistical bias correction, by providing a publicly available software tool that allows users easily to assess the effects / unintended consequences of different "correction" methods. It will be interesting to see whether this makes a substantial change to current practice.*

*I have just three comments:*

1. *If I understand correctly, the tool allows users to assess the effect of bias correction methods on a limited number of threshold-based indices. Some of the visualisations are linked to specific metrics (e.g. the cumulative distribution functions for the spell lengths in Figures 4 and 5). Others, such as the boxplots, are completely generic however. I wonder how easy it would be to link to, say, the xclim library (https://xclim.readthedocs.io/en/stable/) which defines a whole range of other climate indices? If you could import those index definitions and provide some core visualisations - such as boxplots - for them, then ibicus would become a really powerful tool.*

Response: This is correct with one exception: the threshold-based indices within ibicus are meant to be generic, meaning that users can define their metrics and use the full range of generic visualisations on them (ranging from the CDF plots to boxplots and analysis of trend modification). The pointer to the xclim library however is very helpful. In order to minimise dependencies, we decided that explicitly including xclim with the ibicus package is not useful. However, it is easy to export the output of ibicus, compute xclim indices on that and visualise the result of that using xclim. We will include an example of that in the next software release.

2. *Section 4.2.4 claims that bias adjustment changes the uncertainty in an ensemble. Although this kind of claim is commonly made, it is not true: the uncertainty is what it is, and it doesn't change just by massaging the data. Bias adjustment changes the variation which is a symptom of the underlying uncertainty, but that's not the same as changing the uncertainty itself! This is connected to my final point, which is ...*

Response: Thank you for this comment. To address it, as well as the comments of Jorn Van de Velde, we have expanded our discussion of this point and adjusted the text as follows:

Original text: "4.2.4 Evaluation of the uncertainty in the climate model ensemble before and after bias adjustment "

Modified text: "4.2.4 Evaluation of the variation in the climate model ensemble before and after bias adjustment "

Original text: Figure 8 shows that the climate model ensemble spread of the trend in mean seasonal precipitation is modified when applying bias adjustment. This means that not only the trend but also the range of uncertainty and possible worst-case scenarios analysed in impact studies depend on the bias adjustment method used to pre-process the climate model. As shown in the previous sections, the 'best' bias adjustment method for a given use case depends on the variable, region and impact variable studied. The result shown in figure 8 demonstrates that bias adjustment can add an additional source of uncertainty if the method is applied blindly and not evaluated properly. Interestingly the uncertainty range is not necessarily narrowed as has been postulated by some authors (Ehret et al., 2012), but even extended and shifted in some cases.

Modified text: Figure 8 shows that the climate model ensemble spread of the trend of mean seasonal precipitation is modified in different ways by different bias adjustment methods which is in line with previous findings in the literature (Maraun and Widman 2018, Lafferty et al. 2023). Interestingly the variation (often interpreted as the uncertainty range) is not necessarily narrowed as has been postulated by some authors (Ehret et al., 2012), but even extended and shifted in some cases. From this finding, it follows that the range of uncertainty and possible worst-case scenarios analysed in subsequent impact studies might depend on the bias adjustment method used to pre-process the climate model.

The interpretation of this shift in uncertainty is related to the previously discussed questions on trend preservation, namely whether the change in the climate model trend through a statistical bias adjustment method is justified or not. This issue was mentioned by Maraun and Widmann (2018), who discuss that a minimum requirement to justify a change in the uncertainty spread through bias adjustment should be a critical evaluation of the validity of the results and the assumptions of the underlying statistical model. Given the finding in the previous section, namely that the best bias adjustment method depends on the variable, region and impact variable studied, it follows that indiscriminately applying a bias adjustment method across regions and variables without evaluation can shift the spread of the results of subsequent impact studies in a non-justified manner.

*3.      Users may feel as though you've provided a tool that is specifically designed to pull the rug out from under their feet, in the sense that it will almost certainly reveal that there are problems with their chosen bias adjustment method. I am personally rather supportive of any contribution that demonstrates the problems of bias adjustment, but it would perhaps be helpful to provide some constructive suggestions for how to proceed if your software reveals major problems. One such alternative, for example, is to postprocess the entire ensemble within a statistical framework that acknowledges the discrepancies between climate models and the real world, and that aims to derive defensible uncertainty assessments for the real world on the basis of all the available information. There is a fair bit of literature on this: I have made a limited contribution myself, but other authors include Michael Goldstein, Jonathan Rougier, Phil Sansom, Christoph Buser and Claudia Tebaldi (the list goes on!).*

We thank Richard Chandler for the pointer to the literature on post-processing entire ensembles. We have expanded our discussion section quite substantially overall and have added a reference to this strand of literature. However, we felt that a detailed discussion of alternative approaches would be outside the scope of this particular GMD submission.

---

## Author Response (AR1)

Dear Editor,

Please find attached our new manuscript as well as a document highlighting the differences between the previous and new version. We made edits to address the comments by the two reviewers and Richard Chandler which we describe in detail in our reviewer responses posted in the open discussion.

While addressing these comments, we also found an error in the ibicus function for plotting boxplots which excluded some of the locations from being shown in the boxplots. We have corrected this error and updated the boxplots. The additional locations now included in the plots do not change the main results.

Furthermore, we used the matplotlib 'colorblind' palette and tested the colours of our plots for different forms of colour blindness. However, in case the colouring of the plots is not as requested and required by the journal we would welcome support on this issue.

Sincerely,
Fiona Spuler, Jakob Wessel